# Leveraging the Mendelian disorders of the epigenetic machinery to systematically map functional epigenetic variation

Teresa Romeo Luperchio[1†], Leandros Boukas[1,2†], Li Zhang[1], Genay Pilarowski[1‡], Jenny Jiang[1], Allison Kalinousky[1], Kasper D Hansen[1,2]*, Hans T Bjornsson[1,3,4]*

[1]Department of Genetic Medicine, Johns Hopkins University School of Medicine, Baltimore, United States; [2]Department of Biostatistics, Johns Hopkins Bloomberg School of Public Health, Baltimore, United States; [3]Faculty of Medicine, School of Health Sciences, University of Iceland, Reykjavik, Iceland; [4]Landspitali University Hospital, Reykjavik, Iceland

*For correspondence:
khanse10@jhu.edu (KDH);
hbjorns1@jhmi.edu (HTB)

†These authors contributed equally to this work

Present address: ‡Department of Pathology, Stanford University School of Medicine, Stanford, United States

**Abstract** Although each Mendelian Disorder of the Epigenetic Machinery (MDEM) has a different causative gene, there are shared disease manifestations. We hypothesize that this phenotypic convergence is a consequence of shared epigenetic alterations. To identify such shared alterations, we interrogate chromatin (ATAC-seq) and expression (RNA-seq) states in B cells from three MDEM mouse models (Kabuki [KS] type 1 and 2 and Rubinstein-Taybi type 1 [RT1] syndromes). We develop a new approach for the overlap analysis and find extensive overlap primarily localized in gene promoters. We show that disruption of chromatin accessibility at promoters often disrupts down-stream gene expression, and identify 587 loci and 264 genes with shared disruption across all three MDEMs. Subtle expression alterations of multiple, IgA-relevant genes, collectively contribute to IgA deficiency in KS1 and RT1, but not in KS2. We propose that the joint study of MDEMs offers a principled approach for systematically mapping functional epigenetic variation in mammals.

## Introduction

### Mapping causal disease-associated epigenetic variation

A long-standing and fundamental problem in epigenetics is the identification of specific epigenetic changes that *causally* mediate phenotypes through the alteration of transcriptional states. While statistical associations between many diseases/traits and epigenetic changes have been detected, it is typically extremely challenging to rule out the influence of confounders such as the environment, and to determine whether these associations are primary causes vs. secondary consequences (*Rakyan et al., 2011*; *Lappalainen and Greally, 2017*). As a result, to date there are surprisingly few examples of causal relationships between epigenetic alterations and specific phenotypes; notable exceptions include disorders of genomic imprinting (*Barlow and Bartolomei, 2014*), disorders caused by repeat-expansion-induced aberrant promoter hypermethylation (*Sutcliffe et al., 1992*; *LaCroix et al., 2019*), predisposition to some tumor types (*Sakatani et al., 2005*; *Rainier et al., 1993*; *Ogawa et al., 1993*; *Gazzoli et al., 2002*), and – in mice – phenotypes caused by metastable epialleles (*Bertozzi and Ferguson-Smith, 2020*; *Rakyan et al., 2002*).

The recent advent and widespread clinical use of exome sequencing has led to the emergence of a novel class of Mendelian disorders, termed the Mendelian Disorders of the Epigenetic Machinery (MDEMs) (*Fahrner and Bjornsson, 2019*). MDEMs are caused by coding variants disrupting genes

**Figure 1.** The conceptual framework of the present study. (**A**) The causal chain of Mendelian Disorder of the Epigenetic Machinery (MDEM) pathogenesis: the genetic disruption of an epigenetic regulator leads to epigenetic and transcriptomic alterations, which ultimately determine the phenotype. (**B**) We hypothesize that the shared phenotypic features between MDEMs occur because of shared epigenetic and transcriptomic alterations downstream of the genetic disruption of distinct genes. The Venn diagram depicts two MDEMs for convenience, but our approach can be applied to an arbitrary number of MDEMs with shared phenotypes. (**C**) Our approach is designed to derive a list of abnormalities with high probability of causal relevance, by jointly comparing multiple MDEMs. Shown for two MDEMs for convenience. (**D**) Experimental design and workflow for sample generation in our present study. Created with BioRender.com. (**E**) The sample size of our study (number of mice). The ATAC- and RNA-seq samples were generated in parallel (see Materials and methods for details).

The online version of this article includes the following figure supplement(s) for figure 1:

**Figure supplement 1.** Simulation study comparing the ability of the standard approach to detect significant hits shared between experiments to that of our new approach.

encoding for epigenetic regulators, which are generally very intolerant to loss-of-function variation (*Boukas et al., 2019*). This implies the following causal chain underlying MDEM pathogenesis: a coding variant disrupts an epigenetic regulator, leading to downstream epigenomic abnormalities, which in turn give rise to the phenotype, likely by perturbing the transcriptome (*Figure 1A*). As a result, MDEMs may provide a unique lens into the causal relationship between epigenetic/transcriptomic variation and disease. Indeed, studies of Kabuki syndrome type 1 (KS1) – one of the most extensively studied MDEMs to date – have begun unraveling the underpinnings of the neural (*Carosso et al., 2019*), growth (*Fahrner et al., 2019*), cardiac (*Ang et al., 2016*), and immune defects (*Pilarowski et al., 2020*; *Zhang et al., 2015*; *Ortega-Molina et al., 2015*) seen in this disorder.

Here, we leverage MDEMs to design an approach for discovering functionally relevant epigenetic variation, which overcomes limitations such as confounding effects from the environment and reverse causality from the disease process. Our approach is based on a cardinal and thus far unexploited feature of MDEMs, namely their overlapping phenotypic features, despite the causative genetic variants disrupting distinct genes. Such common MDEM features include intellectual disability, growth defects, and immune dysfunction (*Fahrner and Bjornsson, 2019*). We hypothesize that these shared

phenotypes arise because the different primary genetic defects lead to shared downstream epigenomic alterations, which in turn create shared transcriptomic alterations (*Figure 1B*). This hypothesis of a convergent pathogenesis motivates a joint analysis of more than one MDEM, and suggests a simple filter to identify the causal variation at the epigenetic/transcriptomic level: true, disease-driving signals should be detectable in multiple disorders (*Figure 1C*).

## Proof-of-principle: Kabuki syndrome types 1 & 2 and Rubinstein-Taybi syndrome type 1

As proof-of-principle, we implement our proposed approach using mouse models of three MDEMs: two that were clinically indistinguishable prior to the discovery of the underlying genes (Kabuki syndrome types 1 and 2 [KS1 and KS2], caused by haploinsufficiency in histone methyltransferases *KMT2D* and *KDM6A*, respectively), and one that shares phenotypes but is clinically distinct (Rubinstein-Taybi type 1 [RT1], caused by haploinsufficiency in histone acetyltransferase *CREBBP*). Importantly, using mice allows us to: (a) eliminate multiple confounders such as the environment, genetic background, age, and sex, and (b) maintain a consistent sampling of disease-relevant cell types between individuals.

The shared phenotypes of these three syndromes include intellectual disability, growth retardation, and immune dysfunction; the latter is our focus here. In KS1, the immune dysfunction includes hypogammaglobulinemia with low IgA as a consistent feature, as well as abnormal cell maturation which has mostly been characterized in B cells (*Margot et al., 2020*; *Lindsley et al., 2016*). RT1 can also manifest with hypogammaglobulinemia and reduction of mature B cells (*Saettini et al., 2020*.) These defects in KS1 and RT1 are thought to (at least partly) explain the increased susceptibility to infections. In KS2, the immune phenotype has been less extensively studied, in part due to the rarity of the disorder, but there is some evidence of increased infection susceptibility and hypogammaglobulinemia (*Margot et al., 2020*; *Frans et al., 2016*). In mice, the immune phenotypes have been studied in depth only in KS1, where the IgA deficiency closely resembles what is seen in patients with Kabuki syndrome.

Given this potential overlap, we chose to profile positively selected B cells (CD19+) from the peripheral blood of mutant mice, and that of age- and sex-matched wild-type littermates (*Figure 1D*). In order to facilitate a direct comparison of the three MDEMs, we only used female mice, as *Kdm6a* is on the X chromosome (KS2 mouse model), and its complete loss (full knockout) is lethal in male mice.

## Results

### Joint analysis of multiple MDEMs to identify causally relevant epigenetic and transcriptomic variation

The key element of our approach is the joint analysis of the different MDEMs, in order to detect shared molecular (epigenomic/transcriptomic) alterations between them. The simplest statistical methodology for this task would be to perform differential accessibility and expression analyses separately for each disorder, and then obtain a list of the overlapping differential hits. However, this suffers from the major shortcoming that in order to be labeled as differential, a given locus must exceed an arbitrary significance threshold (or rank). When multiple MDEMs are studied, this requirement can lead to severe loss of power and erroneous underestimation of the size of the overlap among the differential hits (*Figure 1—figure supplement 1*; Materials and methods). To avoid this, we recast the problem as testing whether evidence that a set of loci/genes are differential in a given MDEM is informative about the state (null or differential) of the same loci/genes in another MDEM. We show (Materials and methods) that with this formulation, we can use conditional p-value distributions to: (a) estimate the size of the set of overlapping abnormalities and test if it is greater than expected by chance, (b) identify a set of genes that belong to this overlap, and (c) decouple (a) from (b), so that only the identification of specific genes is affected by the multiple testing burden.

### Genome-wide chromatin accessibility profiling reveals extensive overlap between the epigenetic alterations of the three MDEMs

Given that reductions in enzyme activity of KMT2D, KDM6A, and CREBBP are known to alter normal histone modification patterns, we first set out to profile genome-wide chromatin accessibility using ATAC-seq, employing a modified FastATAC protocol (Materials and methods) (*Corces et al., 2016*;

*Buenrostro et al., 2013*). We chose to limit the age range to 2.5–3.5 months as this is the age range we know most about this KS1 model, and this is when the IgA deficiency first manifests in KS1 mice (*Pilarowski et al., 2020*). Starting with a differential accessibility analysis of 7 KS1 vs. 12 wild-type mice (Materials and methods), we discovered 3938 ATAC peaks differentially accessible at the 10 % FDR level. Of these, 1062 (27%) overlapped promoters (defined as ±2 kb from the TSS), and 2876 (73%) were in distal regulatory elements (defined as ATAC peaks outside of promoters).

We then compared KS1 to KS2, focusing on promoters first. We used our new approach in order to detect peaks with shared accessibility disruption. Briefly, we first obtained the 1062 differential promoter peaks from the KS1 vs. WT analysis. Then, we used the distribution of the p-values for these peaks from the KS2 vs. WT analysis to estimate the percentage of shared differential peaks, and individually label each peak as shared or not at the 10 % FDR threshold (see Materials and methods for details). We found that 68.5 % of promoter peaks differentially accessible in KS1 are also differential in KS2 (*Figure 2A*; p < 2.2e-16, 5 KS2 vs. 12 wild-type mice); at the 10 % FDR level, we identified 733 such peaks. For 724 of the 733 (98.8%), accessibility is altered in the same direction in the two syndromes (*Figure 2B*; *Figure 2—source data 1*). Out of these 724 promoter peaks disrupted in both KS1 and KS2, we discovered that approximately 67 % are differential in RT1 as well (*Figure 2C*; p < 2.2e-16, 5 RT1 vs. 7 wild-type mice), again with highly concordant effect sizes (*Figure 2D*).

In total, we identified 420 promoter peaks that show disruption in all three disorders at the 10 % FDR level with concordant effect sizes (*Figure 2—source data 2*). This is >4 times more shared peaks than we find if we perform separate differential analyses and compute the intersection of the resulting differential hits (100 peaks), highlighting that our new approach provides substantial gain in empirical power, as suggested by our simulations. A principal component analysis (PCA) shows that the accessibility signal of these shared disrupted promoter peaks separates each of the three mutant genotypes from their wild-type littermates (*Figure 2e*; *Figure 2—figure supplement 1*). The KS1 mice cluster close to RT1, while KS1 and KS2 cluster separately from each other, with KS2 being closer than KS1 to wild-type, indicating smaller effect sizes of the accessibility alterations. This KS1/2 separation is surprising, since patients have such strong phenotypic overlap that the two syndromes were not considered distinct prior to discovery of the causative genes. However, it should be noted here that, since *Kdm6a* is on the X chromosome, the KS2 female mice are expected to be mosaic with respect to *Kdm6a* knockout, and this may explain the smaller magnitude of their accessibility defects.

Next, we applied the same approach to peaks corresponding to distal regulatory elements. We saw a similar picture, albeit with weaker shared signal (*Figure 2—figure supplement 2*). Specifically, 50.8 % of elements differential in KS1 were estimated as differential in KS2 (p < 2.2e-16), with 815 confidently labeled such elements (10 % FDR). As with promoters, we observed agreement in directionality for the vast majority of peaks (806 out of the 815; *Figure 2—source data 3*). Of the KS1/2 shared elements with concordant directionality, 35.6 % are differential in RT1 (p = 0.0025), yielding a total of 167 shared disrupted distal regulatory elements across the three MDEMs (10 % FDR; *Figure 2—source data 4*). We note that, collectively, the shared hits show a 7.1-fold enrichment at promoters compared to distal elements (Fisher's test, p < 2.2e-16).

Finally, comparing the three MDEMs in a pairwise fashion, we observed that KS1 and KS2 share a greater proportion of their abnormalities than either KS1 or KS2 compared to RT1 (*Figure 2—figure supplement 2*), and verified that this is not driven by the fact that the KS1 and KS2 mice were compared against the same wild-type group (Materials and methods).

## Shared disrupted promoters, but not distal regulatory elements, show bias toward increased accessibility in KS1 and KS2

We explored the direction in which the accessibility of the disrupted peaks changes in mutants compared to wild-type. We found that, at promoters, both the KS1 and KS2 mutants exhibit a substantial shift toward increased accessibility (83.5% and 91.2%, respectively, of significantly disrupted promoter peaks; *Figure 2F*). The same shift is observed at the promoter peaks with shared disruption across the MDEMs (*Figure 2F*), even though in the RT1 mutants the majority (62.2%) of differentially accessible promoter peaks show the opposite pattern, with a shift toward decreased accessibility (*Figure 2F*). In contrast to promoters, disrupted distal regulatory elements in all cases are more evenly split: the percentage with increased accessibility is 41.6 % in KS1, 39.3 % in KS2, and 60 % in RT1 (*Figure 2—figure supplement 2*).

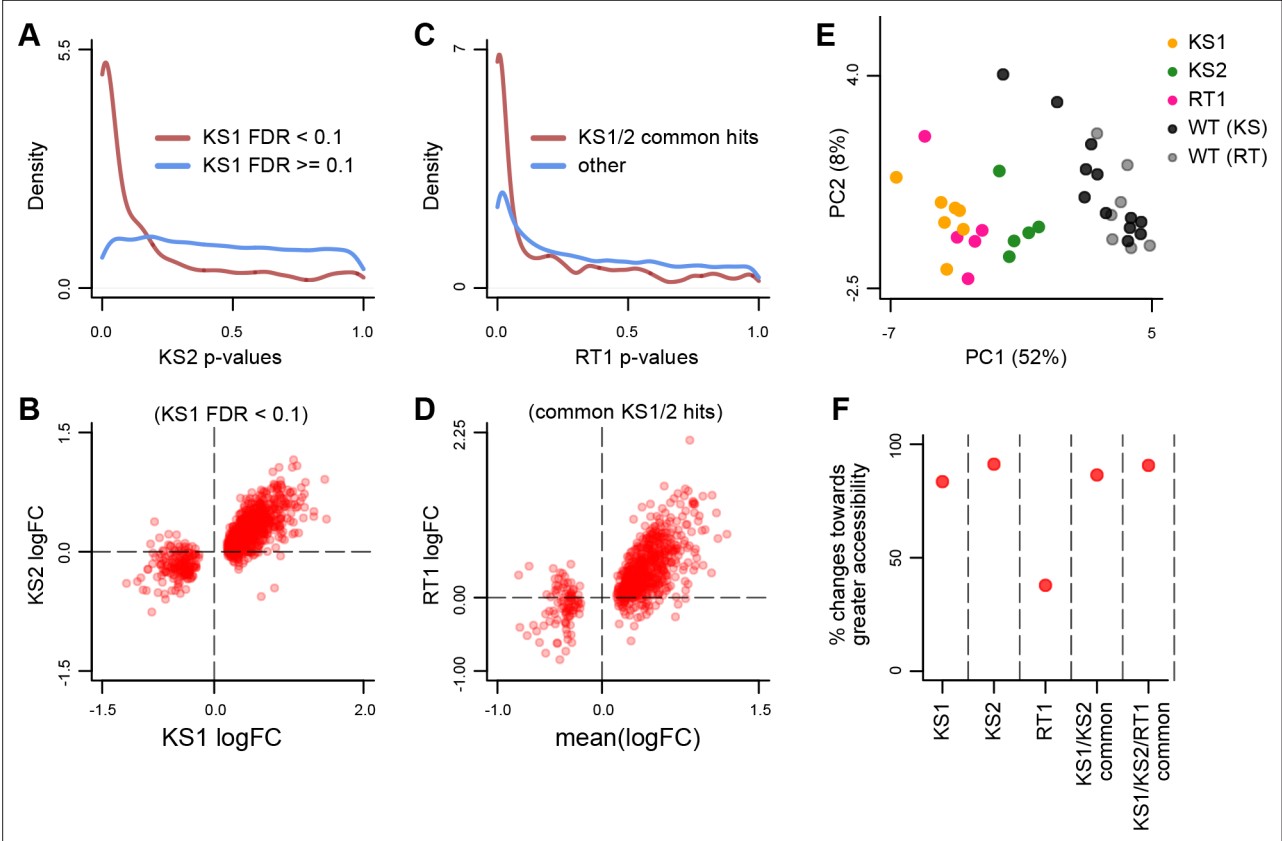

**Figure 2.** Evaluating the overlap between the differentially accessible promoter peaks in Kabuki type 1 (KS1), Kabuki type 2 (KS2), and Rubinstein-Taybi type 1 (RT1) syndromes. (**A**) The distribution of p-values from the KS2 vs. wild-type differential accessibility analysis for promoter peaks, stratified according to whether the same promoter peaks are significantly differentially accessible in the KS1 vs. wild-type analysis (FDR < 0.1; red curve), or not (FDR ≥ 0.1; blue curve). (**B**) Scatterplot of log2(fold changes) from the KS1 vs. wild-type (x-axis) promoter peak differential accessibility analysis against the corresponding log2(fold changes) from the KS2 vs. wild-type analysis (y-axis). Each point corresponds to a peak. Shown are only peaks that are differentially accessible in KS1 (FDR < 0.1). (**C**) The distribution of p-values from the RT1 vs. wild-type differential accessibility analysis for promoter peaks, stratified according to whether the same promoter peaks are shared differentially accessible between KS1 and KS2 (FDR < 0.1, see Materials and methods; red curve), or not (blue curve). (**D**) Scatterplot of log2(fold changes) from the RT1 vs. WT (x-axis) differential accessibility analysis, against the mean log2(fold change) from the KS1 vs. wild-type and KS2 vs. wild-type analyses. Each point corresponds to a peak. Shown are only shared differentially accessible promoter peaks between KS1 and KS2 (FDR < 0.1). (**E**) Principal component analysis plot using only the 420 promoter peaks identified as shared differentially accessible between the three Mendelian Disorders of the Epigenetic Machinery (MDEMs). Each point corresponds to a mouse. (**F**) The proportion of differentially accessible promoter peaks that show increased accessibility in the mutant vs. the wild-type mice.

The online version of this article includes the following figure supplement(s) for figure 2:

**Source data 1.** Coordinates of shared differentially accessible promoter peaks in Kabuki type 1 (KS1) and Kabuki type 2 (KS2) syndromes, along with the corresponding logFC changes.

**Source data 2.** Coordinates of shared differentially accessible promoter peaks in Kabuki type 1 (KS1), Kabuki type 2 (KS2), and Rubinstein-Taybi type 1 (RT1) syndromes, along with the corresponding logFC changes.

**Source data 3.** Coordinates of shared differentially accessible distal regulatory element peaks in Kabuki type 1 (KS1) and Kabuki type 2 (KS2) syndromes, along with the corresponding logFC changes.

**Source data 4.** Coordinates of shared differentially accessible distal regulatory element peaks in Kabuki type 1 (KS1), Kabuki type 2 (KS2), and Rubinstein-Taybi type 1 (RT1) syndromes, along with the corresponding logFC changes.

**Source data 5.** Estimated surrogate variables for the differential accessibility and differential expression analyses.

**Figure supplement 1.** Principal component analysis plots using only the 420 promoter peaks identified as shared differentially accessible between the three Mendelian Disorders of the Epigenetic Machinery (MDEMs).

**Figure supplement 2.** Evaluating the overlap between the differentially accessible distal regulatory elements in Kabuki type 1 (KS1), Kabuki type 2 (KS2), and Rubinstein-Taybi type 1 (RT1) syndromes.

## Transcriptome profiling reveals many expression alterations at genes downstream of promoters with disrupted accessibility

We next interrogated the transcriptome using RNA-seq (Materials and methods) to: (a) test whether the identified epigenetic aberrations in each disorder have direct transcriptional consequences and characterize the latter, and (b) identify the shared expression aberrations across the three disorders, and assess the extent to which these result from shared accessibility aberrations at the associated promoters. To capture both chromatin and transcriptional status at a single time point, we generated the RNA-seq samples in parallel with the samples used for ATAC-seq, from a subset of the same individual mice (Materials and methods). Specifically, we performed RNA-seq on five KS1 mice, five KS2 mice, five RT1 mice, and five and seven wild-type mice from the Kabuki and Rubinstein-Taybi cohorts, respectively.

First, for each disorder, we determined the top differential promoter peaks as ranked by p-value, and estimated the percentage of genes downstream of these promoters that show differential expression; we repeated this by sliding the rank threshold for determining the top peaks from 1000 to 5000. When considering the top 1000 promoter peaks, the percentage of differentially expressed downstream genes is 45.4 % in KS1, 42 % in KS2, and 40 % in RT1 (Materials and methods; *Figure 3— source data 1* contains such genes detected at the 10 % FDR level). In all three syndromes, this percentage gradually drops substantially as the cutoff for labeling a promoter as differentially accessible becomes less stringent (*Figure 3A*), indicating a clear relationship between abnormal promoter accessibility and downstream gene expression dysregulation (with the relationship being noisier in RT1). Emphasizing this relationship, we discovered strong concordance between the direction of abnormal changes at the disrupted promoter-gene pairs: increased or decreased promoter accessibility correlates with increased or decreased gene expression, respectively (*Figure 3B, C and D*; Pearson correlation between promoter accessibility logFC and gene expression logFC = 0.78 for KS1, 0.84 for KS2, and 0.82 for RT1).

Finally, we compared the proportion of differentially expressed genes downstream of the shared disrupted promoter peaks, to the same proportion of genes downstream of the top disrupted promoter peaks unique for each disorder (Materials and methods). We invariably found the genes downstream of the shared disrupted peaks to have a higher chance of dysregulated expression (*Figure 3E*; *Figure 3—figure supplement 1*), supporting our hypothesis that the chromatin alterations at these peaks are more likely to have functional impact.

## A substantial proportion of the shared expression alterations among the three MDEMs arise without concomitant disruption of promoter accessibility

To further dissect the relationship between the shared expression and chromatin abnormalities in the three MDEMs, we sought to define a set of genes with shared expression alterations, without utilizing prior information about the accessibility of their promoter peaks.

Utilizing our method, we discovered high overlap between KS1 and KS2, mirroring the findings at the chromatin level (*Figure 4A, B*). Specifically, we found 397 differentially expressed genes shared between them with concordant direction of effect (10 % FDR; *Figure 4—source data 1*). We then estimated 78.8 % of these to be differential in RT1 (*Figure 4*), resulting in 264 genes shared across the three disorders (10 % FDR), with a preponderance of downregulated genes (*Figure 4E,F*; *Figure 4— figure supplement 1*; *Figure 4—source data 2*; 175 downregulated vs. 89 upregulated genes).

While these 264 genes are significantly enriched in the set of genes with shared disruption of promoter accessibility (p = 0.0001), the magnitude of this enrichment is modest (29 genes in the intersection; odds ratio = 2.37, *Table 1*). The number of genes in the intersection increases to 99 when we also include those harboring shared disrupted regulatory elements nearby (±1 Mb from their promoter peaks). Taken together, these results indicate that there is convergent dysregulation of gene expression in these three MDEMs, which is not always a direct downstream consequence of the shared epigenetic alterations. Nevertheless, in KS1 and KS2 – but not in RT1 – the top differentially expressed genes are more likely to have disrupted promoters than genes further down the differential list (*Figure 4G*).

**Figure 3.** The relationship between differential accessibility of promoter peaks and differential expression of downstream genes in the three Mendelian Disorders of the Epigenetic Machinery (MDEMs). (**A**) The proportion of promoters with differentially expressed downstream genes in Kabuki type 1 (KS1), Kabuki type 2 (KS2), and Rubinstein-Taybi type 1 (RT1) syndromes, estimated for the top ranked differentially accessible promoter peaks. The estimation was repeated for different thresholds for determining the top ranked list. For each MDEM, each point corresponds to a different threshold. Thresholds were slid from 1000 to 5000, in steps of 250. (**B**) Scatterplot of the accessibility log2(fold changes) of differentially accessible promoter peaks, against the expression log2(fold changes) of differentially expressed downstream genes, for each of the three MDEMs. Shown are only pairs where the promoter peak was within the top 1000 differentially accessible promoter peaks (ranked based on p-value), and the downstream gene was differentially expressed (10 % FDR; Materials and methods). Each point corresponds to a gene-promoter pair. In cases where more than one peak in the same promoter was within the top 1000 differentially accessible peaks, the median(log2(fold change)) across all such peaks was calculated. (**C**) and (**D**) An example locus (*Pard3b*) with concordant changes in promoter peak accessibility and downstream gene expression in all three MDEMs. (**E**) The proportion of promoters with differentially expressed downstream genes in KS1, KS2, and RT1, estimated separately for the top uniquely differentially accessible promoters in each MDEM (see Materials and methods), vs. the same proportion estimated for the genes downstream of the 420 shared differentially accessible promoter peaks.

The online version of this article includes the following figure supplement(s) for figure 3:

**Source data 1.** Differentially expressed genes downstream of differentially accessible promoter peaks, along with the corresponding p-values and logFC changes.

**Figure supplement 1.** The distributions of p-values (from the differential expression analyses) for genes downstream of promoters with differentially accessible peaks shared across the disorders, or unique to the particular disorder.

## Integration of transcription factor motifs with chromatin and expression alterations highlights some potentially disrupted regulatory connections

It is well appreciated that chromatin accessibility is intimately linked to transcription factor binding. Accessibility patterns are often established subsequently to recruitment of epigenetic regulators by

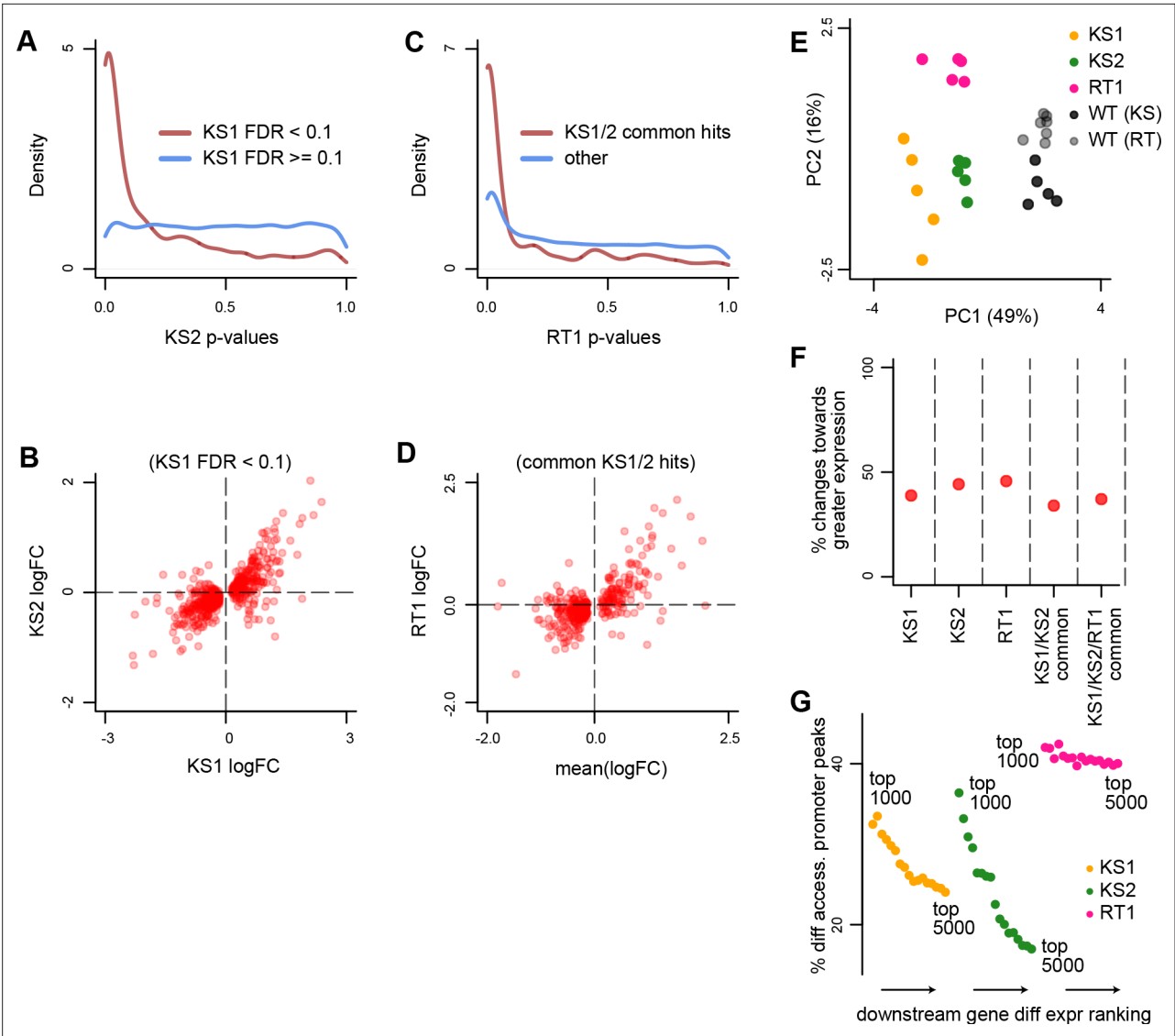

**Figure 4.** Evaluating the overlap between the differentially expressed genes in Kabuki type 1 (KS1), Kabuki type 2 (KS2), and Rubinstein-Taybi type 1 (RT1) syndromes. (**A**) The distribution of p-values from the KS2 vs. wild-type differential expression analysis, stratified according to whether the same genes are significantly differentially expressed in KS1 (FDR < 0.1; red curve), or not (FDR ≥ 0.1; blue curve). (**B**) Scatterplot of log2(fold changes) from the KS1 vs. wild-type differential expression analysis (x-axis), against the corresponding log2(fold changes) from the KS2 vs. wild-type analysis (y-axis). Each point corresponds to a gene. Shown are only genes that are differentially expressed in KS1 (FDR < 0.1). (**C**) The distribution of p-values from the RT1 vs. wild-type differential expression analysis, stratified according to whether the same genes are shared differentially expressed between KS1 and KS2 (FDR < 0.1, see Materials and methods; red curve), or not (blue curve). (**D**) Scatterplot of log2(fold changes) from the RT1 vs. WT (x-axis) differential expression analysis, against the mean log2(fold change) from the KS1 vs. wild-type and KS2 vs. wild-type analyses. Each point corresponds to a gene. Shown are only genes that are shared differentially expressed between KS1 and KS2 (FDR < 0.1). (**E**) Principal component analysis plots using only the 264 genes identified as shared differentially expressed between the three Mendelian Disorders of the Epigenetic Machinery (MDEMs). Each point corresponds to a mouse. (**F**) The proportion of differentially expressed genes that show increased expression in the mutant vs. the wild-type mice. (**G**) The proportion of genes with differentially accessible promoter peaks in KS1, KS2, and RT1, estimated for the top ranked differentially expressed genes. The estimation was repeated for different thresholds for inclusion into the top ranked list. For each MDEM, each point corresponds to a different threshold. Thresholds were varied from 1000 to 5000, in steps of 250.

The online version of this article includes the following figure supplement(s) for figure 4:

**Source data 1.** Shared differentially expressed genes in Kabuki type 1 (KS1) and Kabuki types 2 (KS2) syndromes, along with the corresponding logFC changes.

**Source data 2.** Shared differentially expressed genes in Kabuki type 1 (KS1), Kabuki type 2 (KS2), and Rubinstein-Taybi type 1 (RT1), along with the corresponding logFC changes.

*Figure 4 continued on next page*

*Figure 4 continued*

**Source data 3.** Transcription factor motifs enriched in peaks found in promoters of differentially expressed genes.

**Figure supplement 1.** Principal component analysis plots using only the 264 genes identified as shared differentially expressed between the three Mendelian Disorders of the Epigenetic Machinery (MDEMs).

transcription factors at specific genomic locations (*Voss and Hager, 2014*), while other transcription factors can only bind their cognate motifs if these reside within pre-accessible sites (*John et al., 2011*; *Guertin and Lis, 2010*). We therefore investigated the transcription factor motifs encoded within the differentially accessible peaks in the three disorders, using a set of 233 non-redundant motifs (Materials and methods). We focused on differentially accessible peaks within promoters of differentially expressed genes, reasoning that these are more likely to reveal potentially disrupted regulatory connections with functional relevance. In all three syndromes, we observed only modest enrichments (*Figure 4—source data 3*).

Regulatory wiring disruption can occur not only because of altered motif accessibility, but also theoretically because of abnormal expression of the cognate transcription factors themselves. We thus also performed a search for motif enrichment in promoter peaks corresponding to differentially expressed genes, regardless of whether these peaks are differentially accessible or not. Notable among the significant hits (*Figure 4—source data 3*) are motifs recognized by transcription factors of the NF-kB pathway, which appears affected at the expression level in the three disorders. However, the enrichment effect sizes are again small. Overall, this analysis indicates that the regulatory network disruption in the three MDEMs does not converge to a few dominant transcription factors.

## The collective effect of individually subtle alterations in multiple genes is likely responsible for perturbed IgA production and abnormal B-cell maturation

Our results establish the existence of widespread epigenetic and transcriptional aberrations that are largely shared across the three disorders, suggesting functional relevance. We therefore asked whether these aberrations can explain some specific aspects of the immune dysfunction. We first performed a pathway analysis of the shared disrupted genes (either at the expression or promoter accessibility level; Materials and methods). This yielded several potentially affected pathways (*Figure 5—source data 1* and *Figure 5—source data 2*). However, most of these were of general relevance and did not pinpoint very specific pathologies.

We then reasoned that we might gain more insight by focusing on two of the specific phenotypes seen in KS1: abnormal B-cell maturation and IgA deficiency (*Pilarowski et al., 2020*; *Lindsley et al., 2016*). We set out to test if these are attributable to the collective dysregulation of multiple genes, or to the abnormal expression of a select few. To define relevant gene sets, we first obtained the set of all transcription factors encoded in the mouse genome that are expressed in CD19+ B cells (Materials and methods); this choice was motivated by the fact that transcription factors are critical regulators of cellular differentiation and maturation. We then examined the ranks of these transcription factors in the KS1 p-value distribution and observed a strong shift indicative of global dysregulation (*Figure 5A*; p = 0.001). For IgA deficiency, we assembled a list of 75 genes known to lead to IgA deficiency when individually knocked out in mouse (Materials and methods). Examination of the KS1 p-value ranks of these genes also highlighted a collective shift toward lower p-values (*Figure 5B*; p = 0.03). Together, these results suggest the collective (but often subtle) dysregulation of many genes.

Turning our attention to KS2 and RT1, we observed similar results for transcription factors, with substantial contribution from a set of transcription factors dysregulated in all three MDEMs (*Figure 5A, C*). However, when assessing the IgA deficiency genes, we only observed the signal in RT1, and not in KS2 (*Figure 5B,D*). This was surprising, given the high phenotypic similarity between KS1 and KS2, and prompted us to measure serum IgA in the KS1/2 and wild-type mice (Materials and methods). In agreement with the collective behavior of IgA-related genes, we found no difference in IgA levels between the KS2 and wild-type, while we recapitulated our previous result of IgA deficiency in KS1 mice (*Pilarowski et al., 2020*; *Figure 5E*; p = 0.8 for KS2 vs. WT, p = 0.0008 for KS1 vs. WT, *Figure 5—source data 3*).

**Table 1.** Shared differentially expressed genes with shared differentially accessible promoters in Kabuki type 1, Kabuki type 2, and Rubinstein-Taybi syndromes.

'Up' ('down') indicate increased (decreased) expression or increased (decreased) promoter accessibility in the mutant vs. the wild-type mice. Gene functions were obtained via manual curation.

| Gene name | Gene expression | Promoter accessibility | Gene function |
|---|---|---|---|
| Pard3b | Up | Up | Cell division and cell polarization processes |
| Pbx1 | Up | Up | Transcription factor |
| Epm2a | Up | Up | Serine/threonine/tyrosine phosphatase |
| Zfp365 | Up | Up | Transcription factor |
| Ccdc88a | Up | Up | Actin binding protein |
| Tanc2 | Up | Up | Synaptic scaffolding protein |
| Dip2c | Up | Up | Protein interacting with transcription factors |
| Kif13a | Up | Up | Microtubule-based motor protein |
| Spry2 | Up | Up | Inhibitory activity on receptor tyrosine kinase signaling proteins |
| Ndrg1 | Up | Up | N-myc downregulated gene family member |
| Ebi3 | Down | Down | Interleukin subunit |
| Ppdpf | Down | Down | Regulator of exocrine pancreas development |
| Golim4 | Up | Up | Golgi protein |
| Reln | Up | Up | Secreted extracellular matrix protein |
| Amz1 | Down | Down | Zinc metalloproteinase |
| Slc29a4 | Up | Up | Monoamine transporter |
| Bicd1 | Up | Up | Role in intracellular cargo transport |
| Slc25a4 | Up | Up | Member of the mitochondrial carrier subfamily |
| Nr3c2 | Up | Up | Mineralocorticoid receptor |
| Zfp827 | Up | Up | Transcription factor |
| Slc36a4 | Up | Up | Amino acid transporter |
| Arhgef12 | Up | Up | Guanine exchange factor |
| Tbc1d2b | Up | Up | GTP-ase activating protein |
| Cask | Up | Up | Calcium-calmodulin-dependent serine protein kinase |
| Dmd | Up | Up | Connects cytoskeleton and the extracellular matrix |
| Maged1 | Up | Up | p75 neurotrophin receptor mediated program |
| Chic1 | Up | Up | Cysteine-rich hydrophobic (CHIC) domain containing protein |
| Gprasp1 | Up | Up | G protein-coupled receptor interacting protein |
| Col4a5 | Up | Up | Major collagen of basement membrane |

Finally, we found no evidence that these collective defects in the expression of transcription factors and IgA deficiency associated genes are driven by similar shifts toward abnormal promoter accessibility, with the exception of transcription factor promoters in RT1 (p = 0.04).

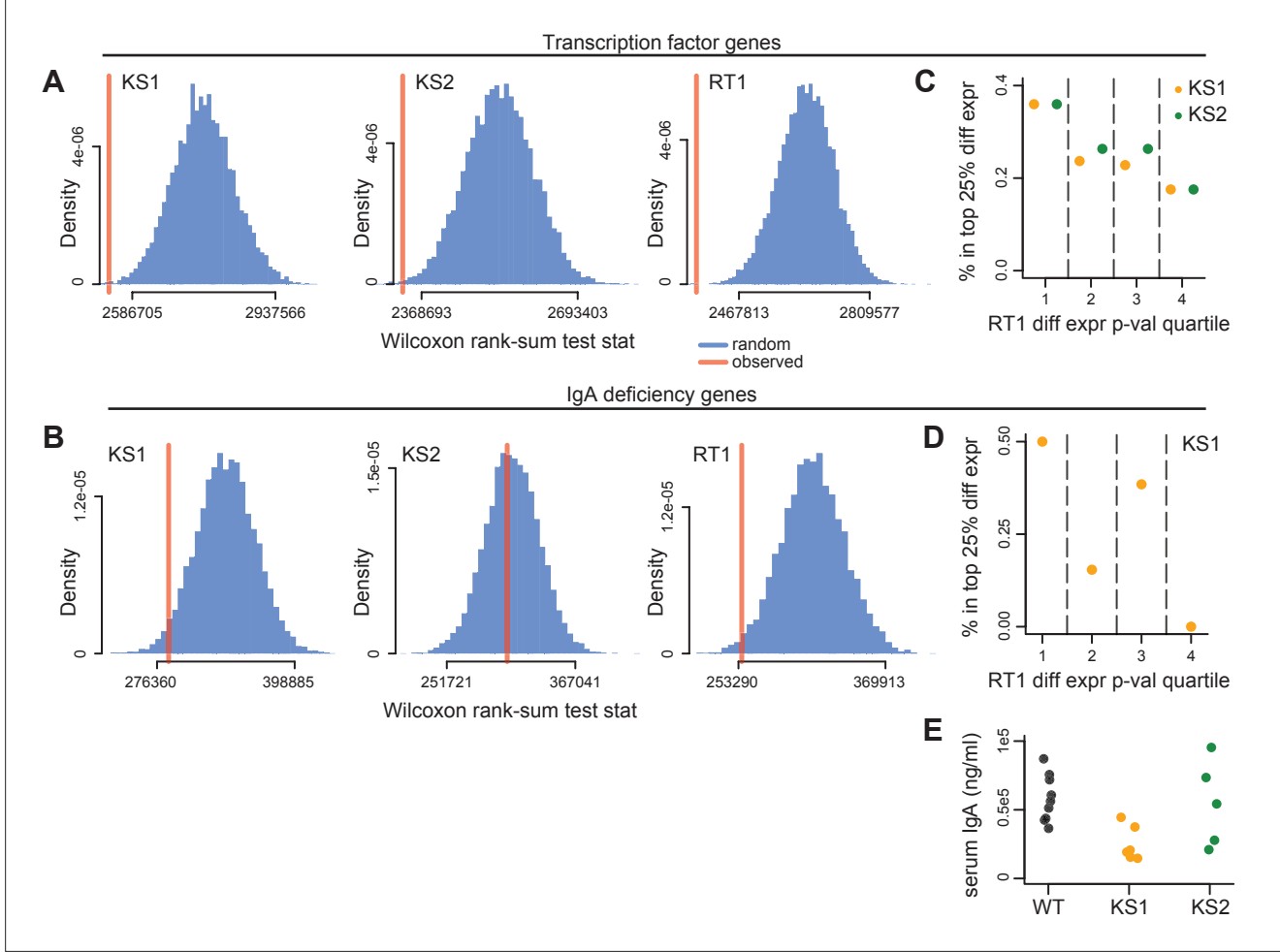

**Figure 5.** Evaluating genes known to encode transcription factors (TFs), or individually contribute to IgA deficiency, for collective expression dysregulation. (**A**) The Wilcoxon rank-sum test statistic (red vertical line) computed after assembling a list of genes encoding TFs expressed in B cells (Materials and methods), and comparing the distribution of their differential expression p-values to the p-value distribution of the rest of the genes included in the differential expression analysis. The blue distribution corresponds to the same statistic computed after randomly sampling gene sets of the same size as TFs, and comparing their p-value distribution to the p-values for the rest of the genes. The resampling was performed 10,000 times. (**B**) Same as (**A**), but for genes known to individually contribute to IgA deficiency (Materials and methods). (**C**) The percentage of TF genes that belong to the top 25 % differentially expressed TFs in Kabuki type 1 syndrome (KS1) (orange dots), and Kabuki type 2 syndrome (KS2) (green dots), stratified according to their p-value quartile in Rubinstein-Taybi type 1 syndrome (RT1). (**D**) Same as (**C**), but for IgA deficiency genes compared in KS1 and RT1. (**E**) Serum IgA levels in KS1, KS2, and wild-type mice.

The online version of this article includes the following figure supplement(s) for figure 5:

**Source data 1.** Top 20 Reactome enriched pathways, using shared differentially expressed genes in Kabuki type 1 (KS1), Kabuki type 2 (KS2), and Rubinstein-Taybi type 1 (RT1) syndromes.

**Source data 2.** Top 20 Reactome enriched pathways, using genes with shared differentially accessible promoters in Kabuki type 1 (KS1), Kabuki type 2 (KS2), and Rubinstein-Taybi type 1 (RT1) syndromes.

**Source data 3.** Serum IgA levels measured in the two types of Kabuki syndrome and wild-type littermates.

# Discussion

Our study is motivated by the hypothesis that the shared phenotypic manifestations seen in MDEMs are attributable to shared underlying epigenomic and transcriptomic abnormalities. We have taken a first step toward validating this hypothesis, by showing that three MDEMs caused by loss-of-function variants in three distinct epigenetic regulators have shared alterations at the chromatin and gene expression level in B cells. However, we have not established that these shared abnormalities are causal for pathogenesis; this will require targeted manipulation of chromatin state and gene expression at

the appropriate cell types and developmental stages. Nonetheless, our results provide some evidence of functionality, as illustrated by the fact that: (a) many chromatin changes at promoters are linked to downstream gene expression changes and (b) systematic expression changes affect genes known to contribute to specific, well-characterized phenotypic features (IgA deficiency, abnormal B-cell maturation) of these MDEMs.

One limitation in our study is the use of only female mice. While it is known that sex differences can influence immune responses and deficiencies (*Klein and Flanagan, 2016*), IgA deficiency has been observed in both male and female KS1 patients, and characterized in both male and female KS1 mice (*Pilarowski et al., 2020*). This supports the notion that our results are relevant to both sexes, although future work is needed to test this for KS2 and RT1. It should also be mentioned that, given the differences in immune system function between mice and humans (*Seok et al., 2013*), some aspects of our results (e.g. some of the disrupted loci/genes) may differ in patients. However, the immune dysfunction in KS1 mice has previously been shown to mimic many aspects of what is seen in patients (*Pilarowski et al., 2020*), and we therefore anticipate that a substantial proportion of the specific changes will be recapitulated. We also expect that the pattern of extensive sharing of abnormalities between MDEMs will hold true.

In terms of understanding the pathogenesis of MDEMs, our findings clearly point toward a generalized, systems-level dysregulation, with a multitude of cellular processes/pathways affected. This is supported both by the extensive sharing of chromatin and expression alterations between the three disorders, and by the several Reactome pathways that appear affected. From our present study it is unclear how exactly these combine to ultimately give rise to the phenotypic manifestations; elucidating this will be an important challenge going forward. It is also worth noting that the emergent picture bears similarities to the molecular basis of complex diseases. This is perhaps not unexpected, given that epigenetic regulators are typically *trans*-acting proteins that act at many locations. It also suggests that, even though MDEMs are single-gene Mendelian disorders with respect to their inheritance pattern, when it comes to their underlying molecular pathogenesis they might best be conceptualized as effectively complex disorders, with many widely distributed, small-effect perturbations, ultimately generating the phenotype (*Boyle et al., 2017*). This may also explain the broadness of the phenotype in MDEMs (*Bjornsson, 2015*), and the decreased penetrance of many phenotypes in patients that are fully penetrant in mouse models.

It is notable that we find greater molecular overlap between KS1 and KS2 than between either of them and RT1, in agreement with the greater similarity between the two KS types at the phenotypic level. It should be mentioned, however, that specific sub-phenotypes provide exceptions to this rule, as evidenced by the abnormalities in IgA deficiency genes, which are present in KS1 and RT1 but are absent in KS2. Together, these results suggest that deep phenotyping of MDEMs combined with molecular characterization in disease-relevant cell types can yield new insights into the pattern of their shared features. We also note that, for a complete understanding of each MDEM individually, our cross-MDEM comparison approach should ultimately be complemented by disorder-specific analyses, as some disrupted loci/genes/pathways may show disorder-specific abnormalities.

One unexpected finding was that, at promoters, almost all of the shared disrupted peaks exhibit a shift toward a more open chromatin state, even though the causative mutations of all three disorders would theoretically be expected to push toward a more closed chromatin state, based on the specific histone marks they are thought to affect (*Fahrner and Bjornsson, 2014*). One possible explanation is that these shared hits represent indirect effects, arising downstream of the initial effects of the mutations. Alternatively, the hypothesis that loss of the epigenetic regulators disrupted in our three disorders would lead to closed chromatin may not hold. Finally, there is the possibility that the causative mutations lead to a non-specific cellular compensatory response, which causes increased chromatin openness at several genomic locations such as the adaptive stress response (*Brose et al., 2012*). The latter is supported by the fact that many of the shared genes uncovered in this study are not known to be directly targeted by the KMT2D, KDM6A, or CREBBP proteins. Regardless of the exact reason, this observation warrants future exploration.

We note that our study differs from recent studies of DNA methylation in the peripheral blood of MDEM patients (*Aref-Eshghi et al., 2020*; *Sobreira et al., 2017*; *Butcher et al., 2017*). In these studies, the goal is to derive 'episignatures' with the capacity for robust phenotypic prediction. As a result, these episignatures include a set of CpGs that jointly maximize the ability to separately classify

individuals with a given MDEM from controls, without regard to the causal role (if any) of these CpGs in disease pathogenesis. While this does not limit their potential usefulness, and does not exclude the possibility that changes in the methylation state of some of these CpGs may be functionally related to disease pathogenesis, our strategy is specifically designed to yield a catalog of abnormalities with primary functional role in shared MDEM pathogenesis.

In summary, we propose the study of the MDEMs as a principled approach for systematically mapping causally relevant epigenetic variation in mammals. The shared hits among the three MDEMs studied here almost exclusively demonstrate an increase in open chromatin at promoters, which is counterintuitive to the function of the individual causative genes and may either suggest a previously unexpected role for them or an undescribed systemic compensatory response. Finally, we suggest that MDEMs are effectively complex disorders with respect to their molecular pathogenesis, arising from widely distributed epigenetic perturbations across the genome.

# Materials and methods

## Key resources table

| Reagent type (species) or resource | Designation | Source or reference | Identifiers | Additional information |
|---|---|---|---|---|
| Strain, strain background (*Mus musculus*, both sexes) | *Kmt2d*$^{+/\beta Geo}$ mice (fully backcrossed to C57BL/6J) | Originally from Bay Genomics and described in PMID:2527309625273096. | *Kmt2d*$^{+/\beta Geo}$, *Mll2*$^{Gt(RRt024)Byg}$ RRID:MGI:5829565 | A previously characterized mouse model of Kabuki syndrome (type 1). |
| Strain, strain background (*Mus musculus*, females only) | *Kdm6a*$^{\pm}$ mice (fully backcrossed to C57BL/6J, observed male lethality) | Ordered from EMMA (European Mouse Mutant Archive) | *Kdm6a*$^{+/}$, *Kdm6a*$^{tm1d(EUCOMM)Wtsi}$ MGI:4434460 | A previously characterized mouse model of Kabuki syndrome (type 2). Transition from *Kdm6a*$^{tm1a(EUCOMM)Wtsi}$ to *Kdm6a*$^{tm1d(EUCOMM)Wtsi}$ performed in Bjornsson laboratory. |
| Strain, strain background (*Mus musculus*, both sexes) | *Crebbp*$^{\pm}$ mice (fully backcrossed to C57BL/6J) | Ordered from Jackson laboratory and described in PMID:10673499 | *Crebbp*$^{+/-}$ *Crebbp*$^{tm1Dli}$, RRID:MGI:2175793 | A previously characterized mouse model of Rubinstein-Taybi syndrome (type 1). |
| Sequence-based reagent | βGeo F | This paper | PCR primers | CAAATGGCGATTACCGTTGA |
| Sequence-based reagent | βGeo R | This paper | PCR primers | TGCCCAGTCATAGCCGAATA |
| Sequence-based reagent | Tcrd (control) F | This paper | PCR primers | CAAATGTTGCTTGTCTGGTG |
| Sequence-based reagent | Tcrd (control) R | This paper | PCR primers | GTCAGTCGAGTGCACAGTTT |
| Sequence-based reagent | Kdm6aTm1c F | This paper | PCR primers | AAGGCGCATAACGATACCAC |
| Sequence-based reagent | Kdm6aTm1c, Floxed LR | This paper | PCR primers | ACTGATGGCGAGCTCAGACC |
| Sequence-based reagent | Tcrd (control) F- | This paper | PCR primers | CAAATGTTGCTTGTCTGGTG |
| Sequence-based reagent | Tcrd (control) R | This paper | PCR primers | GTCAGTCGAGTGCACAGTTT |
| Sequence-based reagent | *Crebbp* R-T F | This paper | PCR primers | TAAGCAGCAGCATCCTTTGG |
| Sequence-based reagent | *Crebbp* R-T_WT | This paper | PCR primers | CCTGACAATGTGTCATGTGAT |
| Sequence-based reagent | *Crebbp* R_T_MUT R: | This paper | PCR primers | ATGCTCCAGACTGCCTTGGGA |
| Commercial assay or kit | IgA ELISA kit | Thermo | Catalog # EMIGA | |

*Continued on next page*

*Continued*

| | | | | | |
|---|---|---|---|---|---|
| Commercial assay or kit | CD19 positive selection | Miltenyi | 130-052-201 | | |
| Commercial assay or kit | Tagmentation | Illumina Nextera | FC-121–1030 | | |
| Commercial assay or kit | Digitonin | Promega | G9441 | | |
| Commercial assay or kit | DNA clean and concentration kit | Zymo | D4013 | | |
| Commercial assay or kit | Select A size purification | Zymo | D4080 | | |
| Commercial assay or kit | DNA high sensitivity | Agilent | 5067–4626 | | |
| Commercial assay or kit | Qubit dsDNA HS | Thermo | Q32851 | | |
| Commercial assay or kit | Direct-zol RNA microprep | Zymo | R2060 | | |
| Commercial assay or kit | Quant-iT RiboGreen | Thermo | R11490 | | |
| Commercial assay or kit | RNA HS Assay kit | Thermo | Q32852 | | |
| Commercial assay or kit | RNA 6000 Pico | Agilent | 5067–1513 | | |
| Commercial assay or kit | NEBNext Poly(A) mRNA isolation module | New England Biolabs | E7490 | | |
| Commercial assay or kit | NEBNext Ultra II Directional RNA Library Prep kit | New England Biolabs | E7760/E7765 | | |
| Commercial assay or kit | KAPA library Quantification kit | KAPA | KK4824 | | |
| Software, algorithm | BowTie2 | PMID: 22388286 | RRID:SCR_016368 | Default parameters | |
| Software, algorithm | Samtools | PMID:19505943 | RRID:SCR_002105 | | |
| Software, algorithm | MACS2 | PMID: 18798982 | RRID:SCR_013291 | Keep-dup = all | |
| Software, algorithm | DESeq2 | PMID: 25516281 | RRID:SCR_015687 | | |
| Software, algorithm | Surrogate Variable Analysis | PMID: 17907809 | RRID:SCR_012836 | | |
| Software, algorithm | Salmon | PMID: 28263959 | V0.10 RRID:SCR_017036 | | |
| Other | GEO submission of all data | Accession GSE162181 | RRID:SCR_005012 | | |

## Mice

We performed all mouse experiments in accordance with the National Institutes of Health Guide for the Care and Use of Laboratory Animals and all were approved by the Animal Care and Use Committee of the Johns Hopkins University (protocol number: MO18M112). For all comparisons, we used wild-type littermates as controls for each batch, although we did pool wild-type animals in some analyses. Numbers, outcome measures, and statistical testing are described throughout the text as well as in the transparent reporting form.

We genotyped mice using standard genotyping and PCR methods. For all comparisons, we used wild-type littermates.

*KS1. Kmt2d$^{+/\beta Geo}$* mice are fully backcrossed to C57BL/6 J and this backcrossing is verified by SNP genotyping (**Bjornsson et al., 2014**). These mice are also known as *Mll2$^{Gt(RRt024)Byg}$*. They were originally obtained from BayGenomics and were fully backcrossed in the Bjornsson laboratory.

Primers:

> βGeo F-CAAATGGCGATTACCGTTGA, R-TGCCCAGTCATAGCCGAATA;
> Tcrd (control) F-CAAATGTTGCTTGTCTGGTG, R-GTCAGTCGAGTGCACAGTTT

*KS2. Kdm6a$^{\pm}$* mice were acquired from European Mouse Mutant Archive (EMMA) but this model has also been called: *Kdm6a$^{tm1a(EUCOMM)Wtsi}$*. Mice were crossed with flippase expressing mice (B6.Cg-T-g(ACTFLPe)9205Dym/J) (Jackson Laboratories) to remove the third exon of *Kdm6a*, and then progeny were crossed with Cre expressing mice driven by CMV (B6.C-Tg(CMV-cre)1Cgn/J) (Jackson Laboratories), to generate the *Kdm6a$^{tm1d(EUCOMM)Wtsi}$* allele. Mice were backcrossed on C57BL/6 J to maintain the *Kdm6a$^{tm1d(EUCOMM)Wtsi}$* allele.

Primers:

> Kdm6aTm1c F-AAGGCGCATAACGATACCAC, Floxed LR-ACTGATGGCGAGCTCAGACC;
> Tcrd (control) F-CAAATGTTGCTTGTCTGGTG, R-GTCAGTCGAGTGCACAGTTT

*RT1. Crebbp$^{\pm}$* mice, also known as *Crebbp$^{tm1Dli}$*, were acquired from Jackson laboratories but established by **Kung et al., 2000**. These mice were maintained on a C57BL/6 J background in the Bjornsson laboratory.

Primers:

> R-T F: TAAGCAGCAGCATCCTTTGG, R-T_WT R: CCTGACAATGTGTCATGTGAT, R_T_MUT R:
> ATGCTCCAGACTGCCTTGGGA

For this study, multiple mating pairs of wild-type mice crossed with heterozygous mutants were used. The resulting litters from these mate pairs were a mix of mutant and wild-type offspring, all sharing the same genetic background. The wild-type controls that we use in our differential analyses are therefore either siblings to the mutants or cousins (because we have many litters of mice) within each disease model cohort. This ensures our study is not confounded by systematic differences in genetic background between wild-type and mutants.

## Sex disaggregation

We performed all experiments in female mice to enable a comparison between all three disease models, as *Kdm6a* (KS2 model) is present on the X chromosome, and its full knockout is in our experience uniformly lethal in male mice. Therefore, we are unable to present sex-disaggregated data.

## Testing for statistically significant overlap between two lists of differential features and identifying the common hits

Our problem is cast in the following setting. Assume we have performed two experiments, each of which involves measuring multiple features (e.g. genes or peaks) in two conditions and performing a differential analysis. The two experiments measure the same set of features. Because the two experiments investigate different biological systems, we do not expect the set of (true) differential features to be identical. But we are interested in the extent of the overlap between the two sets of features, specifically: (a) Is there statistically significant overlap between the two sets of differential features and how big is it? (b) Which features are differential in both lists?

Our approach to these questions is a conditional approach: we ask, does information about the result in experiment 1 affect our interpretation of experiment 2?

We first (arbitrarily) designate one of the two experiments as experiment 1. We test $m$ features, and for each feature , we let $X_i$ be a factor with values in {0,1} expressing whether the feature was significantly differential in experiment 1 ($X_i = 1$), or not ($X_i = 0$). We are interested in whether the variable $X = (X_1, \ldots, X_m)$ is an informative covariate for experiment 2, using terminology from recent work in covariate-powered multiple hypothesis testing (**Ignatiadis et al., 2016**; **Chen et al., 2021**).

We now consider experiment 2. We split the features into two groups, conditional on the results in experiment 1. Group 1 consists of the features which were found to be differential in experiment 1

(the size of group 1 is $n$) and group 0 consists of the features which were not differential in experiment 1 (of which we have $m - n$). We let each group have its own proportion of differential features, that is, we introduce parameters $\pi_{1|0}$ and $\pi_{1|1}$ . Let $Y_i$ be the indicator whether the  th feature is differential in experiment 2 or not. Then

$$P\left(Y_i = 1\right) = P\left(Y_i = 1 | X_i = 0\right) P\left(X_i = 0\right) + P\left(Y_i = 1 | X_i = 1\right) P\left(X_i = 1\right)$$

$$\pi_{1|0} P\left(X_i = 0\right) + \pi_{1|1} P\left(X_i = 1\right)$$

Our null hypothesis is that experiment 1 is not informative about experiment 2, or in other words

$$H_0 : \pi_{1|0} = \pi_{1|1}$$

Under this null hypothesis $P\left(Y_i = 1\right) = \pi_{1|1}$ . Furthermore, an estimate of $\pi_{1|1}$ should have the same distribution as the proportion of significant features in a random sample of $n$ features from experiment 2, which we term $\hat{\pi}_1^{(n)}$ .

This gives us the following method for testing $H_0$ :

1. Analyze experiment 1 and decide which features are significantly differential or not.
2. Analyze experiment 2, but only the features which were called differential in experiment 1 to estimate $\hat{\pi_{1|1}}$ .
3. Repeatedly, draw $n$ features and estimate $\hat{\pi}_1^{(n)}$ to get a null distribution.

In practice, we can estimate $\hat{\pi_{1|1}}$ and $\hat{\pi}_1^{(n)}$ using a number of different methods that produce estimates of the proportion of true null hypotheses (and thus, of the proportion of false null hypotheses, our statistic of interest here) among a set of hypotheses tested. We here used Storey's method as implemented in the qvalue package in R (**Storey, 2003a**; **Storey and Tibshirani, 2003b**), with the 'pi0.method' parameter set to 'bootstrap'. This tells us the size of the overlap and the extent to which it is significantly greater than what expected by chance. To estimate which features are in the overlap, we use the qvalue() function on the features which are significant in the analysis of experiment 1, with the FDR level set to 10 %.

## Simulation study

We performed a simple simulation study, to test if our method leads to increased ability to detect overlap between lists of differential features, in a setup with known ground truth. We simulated three experiments as follows. Each experiment consisted of testing 10,000 features between two groups. Of these 10,000 features, there were 2000 true differential features and 8000 true null features. There were 1400 true shared differential features between the first two experiments, and 1000 shared true differential features between all three experiments. For each experiment, the null features were simulated from a normal distribution with mean = 0 and variance = 1, for both groups. The differential features were simulated from a normal distribution with mean = 0 and variance = 1 in one group, and mean = 0.5 and variance = 1 in the other. The sample size was always fixed at 75. For each feature, we performed a two-sample t-test, using the col_t_welch() function from the matrixTests R package.

We then estimated the overlap between the experiments using either our method (as described in the previous section) or the standard approach, which consisted of separately identifying significant features at the 10 % FDR level in each experiment, and then counting the number of features in the intersection across experiments. We repeated the simulation 1000 times. Our results confirmed that our method provides substantially greater ability to estimate the size of the overlap (**Figure 1—figure supplement 1**). In addition, our method had an average proportion of false discoveries (i.e. features labeled as belonging to the overlap that were not truly differential across the experiments) equal to 10.9%, very close to the nominal FDR of 10 %.

## Blood cell isolation

We obtained peripheral blood from 2.5- to 3.5 -month-old female mice by facial vein bleed. 150–250 µL blood was collected in $K_2$EDTA blood collection tubes (BD Microtainer 365974) and red blood cells were lysed for 7–15 min at room temperature in 2 mL red blood cell lysis solution (15.5 mM NH4Cl,

1 mM KHCO3, 0.01 mM EDTA). We diluted lysed blood with excess balanced salt solution (Gey's or 1× PBS), manually removed large clots using pipet tip, and spun at 500 g for 10 min 4'. Second lysis at room temperature was performed for samples with large amounts of remaining red blood cells then spun. We isolated CD19+ B cells by positive selection using CD19+ microbeads for mouse (Miltenyi 130-052-201) following manufacturer's protocols, then counted and aliquoted samples on ice to further process for ATAC-seq and RNA-seq. We note that each RNA-seq sample was an aliquot of the same cell harvest as the ATAC-seq sample from the same mouse (though for some mice we only performed ATAC-seq and not RNA-seq).

## ATAC-seq

We performed ATAC-seq using a modified FastATAC protocol (*Corces et al., 2016*; *Buenrostro et al., 2013*). Specifically, we resuspended 5 k cells per reaction in 1× PBS and quickly spun to remove residual EDTA from isolation steps, and then resuspended in tagmentation reaction mix for 30 min (2.5 µL TD1, 1 × TD Buffer, Illumina Nextera DNA, FC-121–1030; 0.25 µL 1 % digitonin, Promega G9441; 1× PBS;) gently shaking (300 rpm on Eppendorf thermomixer) at 37'. We purified reactions using Zymo DNA Clean and Concentrator-5 kit (Zymo D4013) following manufacturer's protocols and eluted with 10.5 µL water to recover 10 µL. Each reaction was then amplified and indexed as described (*Corces et al., 2016*); total sample amplification cycles range from 6 to 10 cycles. After indexing and amplification, we purified samples using Select-A-Size purification columns (Zymo D4080) with a cutoff of 150 bp to remove adapter dimers to allow for efficient sequencing on patterned flow cells, checked library size on BioAnalyzer using DNA High Sensitivity reagents (Agilent 5067–4626) and determined concentration using Qubit dsDNA HS Assay Kit (ThermoFisher Q32851). We pooled and sequenced on Illumina HiSeq4000 using PE flow cells with 100-8-8-100 read length using standard manufacturer's protocols. Samples were clustered to aim for 60 M reads per sample Samples were demultiplexed using Illumina pipeline bcl2fastq2 v2.20 with all defaults except `--use-bases-mask` Y100n, I8, I8, Y100n.

## ATAC-seq mapping and peak calling

We mapped the ATAC-seq reads to the mm10 mouse assembly using bowtie2 (*Langmead and Salzberg, 2012*), with default parameters. We removed duplicate reads with the 'MarkDuplicates' function from Picard (RRID:SCR_006525; version 2.23.8;http://broadinstitute.github.io/picard/), and subsequently also removed mitochondrial reads using samtools (*Li et al., 2009*). We then created genotype-specific meta-samples, by merging all the individual bam files corresponding to samples from mice of a given genotype. This yielded one meta-sample for KS1, one for KS2, and one for RT1. For wild-type mice, we created two such meta-samples, one from the wild-type littermates of the KS1 and KS2 cohorts (to which the KS1 and KS2 mutant mice were compared to), and one for the wild-type littermates of the RT1 cohort (to which the RT1 mutant mice were compared to). For each of the five resulting meta-samples, we then called peaks using MACS2 (*Zhang et al., 2008*), with the 'keep-dup' parameter equal to 'all'.

## ATAC-seq differential analysis

We first defined the set of features to be tested as differential, by unionizing the peaks from all meta-samples. After excluding intervals overlapping ENCODE blacklisted regions (*Amemiya et al., 2019*), we obtained 78,193 genomic intervals (median size = 690 bp, 95th percentile = 1774, range = 151–11,363). To verify that these intervals are not likely to be false positives, we compared them to publicly available DNase Hypersensitivity Sites in B cells (CD19+) from the ENCODE project (https://www.encodeproject.org/experiments/ENCSR000CMM/). We converted the DHS coordinates from mm9 to mm10 using liftOver. We then unionized the intervals from the two DHS replicates to create a common set of 112,728 DHSs. We found that 78,101 of our 78,193 regions (99.88%) overlapped DHSs, providing strong orthogonal evidence that they represent true B-cell regulatory regions.

We then counted the number of reads from each sample that map to each of the 78,193 features, using the featureCounts() function from the Rsubread R package (*Liao et al., 2019*), with the following parameters: requireBothEndsMapped = TRUE, countChimericFragments = FALSE, countMultiMappingReads = FALSE, minOverlap = 3. This resulted in a count matrix with rows corresponding to features (the aforementioned genomic intervals), and columns to samples. This count matrix served

as input for the differential analysis, which we performed using DESeq2 (*Love et al., 2014*). We only retained features with a median (across samples) count greater than 10; this filter resulted in 71,651 features for the KS1 vs. wild-type analysis, 73,189 features for the KS2 vs. wild-type analysis, and 62,386 features for the RT1 vs. wild-type analysis. We used Surrogate Variable Analysis (*Leek and Storey, 2007*) to estimate unobserved confounding variables, and adjusted for those in the differential analysis (without explicitly including other covariates in the model; *Figure 2—source data 5*).

To derive the list of features overlapping promoters, we first obtained promoter coordinates with the promoters() function from the EnsDb.Mmusculus.v79 R package, with the parameters 'upstream' and 'downstream' both equal to 2000. We subsequently restricted to protein-coding transcripts, using the 'tx_biotype' filter. The overlapping features were then obtained using the findOverlaps() function from the GenomicRanges R package.

## RNA-seq

We spun approximately 100 –500k cells at 300–500 g for 5 min at 4', homogenized in Trizol (Invitrogen 15596018) and stored at –80°C until extraction. We extracted and isolated RNA by phase separation using standard protocols followed by purification using the Direct-zol RNA microprep kit (Zymo R2060) with an on-column DNAse step per manufacturer's directions. Once purified, we quantified RNA using Quant-iT RiboGreen RNA Assay Kit (ThermoFisher R11490) or Qubit RNA HS Assay Kit (ThermoFisher Q32852), and checked quality by Bioanalyzer with RNA 6000 Pico Kit (Agilent 5067–1513). All samples show high-quality RNA with RIN >9. We used 20 ng RNA per KS1 and KS2 and matched wild-type sample and 100 ng per RT and matched wild-type sample as input to capture mRNA (NEBNext Poly(A) mRNA Magnetic Isolation Module; NEB #E7490) followed by library generation using NEBNext Ultra II Directional RNA Library Prep Kit for Illumina (NEB E7760/E7765) per manufacturer's protocols. We determined library size and quality using BioAnalyzer with DNA High Sensitivity reagents (Agilent 5067–4626), and determined concentration using Qubit dsDNA HS Assay Kit (ThermoFisher Q32851) and KAPA Library Quantification Kit for qPCR (KAPA KK4824). We pooled samples and sequenced on Illumina HiSeq4000 using PE flow cells with 100-8-8-100 read length using standard manufacturer's protocols. Samples were clustered to aim for 60 M reads per sample. Samples were demultiplexed using Illumina pipeline bcl2fastq2 v2.20.

## RNA-seq mapping and differential analysis

We first obtained a FASTA file (Mus_musculus.GRCm38.cdna.all.fa.gz) containing all mouse cDNA sequences from Ensembl (http://uswest.ensembl.org/Mus_musculus/Info/Index, version 91, downloaded January 2018). We used this file to build an index and pseudo-map the RNA-seq reads with Salmon (v0.10) (*Patro et al., 2017*). We subsequently imported the resulting transcript quantifications into R to get gene-level counts, using the tximport R package (*Soneson et al., 2016*). The differential analysis was then performed with DESeq2, following the same steps as with ATAC-seq. The exclusion of genes with median count across samples ≤10 resulted in 12,566 genes tested in the KS1 vs. wild-type analysis, 12,529 genes tested in the KS2 vs. wild-type analysis, and 12,537 genes tested in the RT1 vs. wild-type analysis.

## Principal component analysis

All PCA plots were generated as follows. We first applied a variance stabilizing transformation to the count matrices (either genes-by-samples or genomic-intervals-by-samples), as implemented in the vst() function from DESeq2. We then used the resulting matrix to perform the PCA with the plotPCA() function.

## Pairwise comparisons between the disorders

We identified greater overlap between the differentially accessible regions identified in KS1 and KS2, than between the differentially accessible regions identified in either of KS1 or KS2 and RT1. To verify that this is not driven by the fact that KS1 and KS2 were compared against the same wild-type group, we re-estimated the overlap, after first conducting differential analyses where KS1 and KS2 mice were compared to separate wild-type cohorts (eight wild-type mice for the KS1 cohort and four mice for the KS2 cohort, respectively). This again revealed the same picture: 73.8 % of differentially accessible regions (promoters or distal regulatory elements) in KS1 are estimated to be differential in KS2 as well,

whereas only 23.1 % of differentially accessible regions (promoters or distal regulatory elements) in RT1 are estimated as differential in KS2.

## Identification of differentially expressed genes with differentially accessible promoter peaks

For *Figure 3B*, we first selected the genes downstream of the top 1000 differentially accessible promoter peaks, the latter being ranked based on their p-values in each disorder. Out of these genes, we retained those differentially expressed using the qvalue() function from the qvalue R package, with the gene p-values as input and the 'fdr.level' parameter set to 0.1. In cases where there were more than one peak in the same promoter, we calculated the median logFC across these peaks. For *Figure 3A*, we slid the rank threshold for determining the top differentially accessible promoter peaks in each disorder from 1000 to 5000, and estimated that the proportion of differentially expressed downstream genes used the pi0est() function from the qvalue R package with the 'pi0.method' parameter set to 'bootstrap'.

For *Figure 3E*, the uniquely differentially accessible promoter peaks for each disorder were defined as peaks that are ranked within the top 1000 for that disorder (based on p-value), but that are not shared across all three disorders, and are not ranked within the top 5000 peaks for any of the other two disorders. This subsetting, as well as the subsetting for shared peaks, results in a relatively small number of p-values (between 174 and 572). Since it is known that the estimation of the proportion of truly differential features can be unreliable in such cases, we also visually inspected the corresponding p-value distributions (*Figure 3—figure supplement 1*). The main issue is the choice of the $\lambda$ tuning parameter, which defines a cutoff above which p-values are assumed to come from truly non-differential features. The qvalue() function provides two options for automatically choosing $\lambda$, the 'bootstrap' (which we use throughout this study) and the 'smoother', both of which are based on heuristic procedures.

In the case of KS1, we observed that the proportion of differentially expressed genes estimated by either the bootstrap or the smoother procedure does not agree with the behavior of the p-value distributions. Specifically, the p-value distribution exhibits somewhat greater concentration close to 0 for the genes downstream of the shared differentially accessible promoter peaks (*Figure 1—figure supplement 1a*; 10th percentile = 0.0003 vs. 0.001). In contrast, the estimate with the bootstrap procedure is 68.4 % for genes downstream of the uniquely differentially accessible promoter peaks and 42.5 % for genes downstream of the shared differentially accessible promoter peaks. The corresponding estimates with the smoother procedure are even more extreme: 75% and 18.2%, respectively. We reasoned that this discordance between the estimates and the behavior of the distributions is likely because the p-value distribution for genes downstream of the uniquely differentially accessible promoter peaks exhibits a bump in the middle (*Figure 3—figure supplement 1A*). Reasoning that this bump is unlikely to be caused by p-values corresponding to truly non-null genes, we manually set $\lambda$ equal to 0.5, that is, we assumed that p-values greater than 0.5 come from truly non-differential genes. This yielded an estimate of the proportion of truly differential genes equal to 34.5 %. Similar estimates are yielded for $\lambda$ values in the range 0.3–0.5 (min estimate = 29.5%, max estimate = 34.5%). For the genes downstream of the shared differentially accessible peaks, this range of $\lambda$ values produces estimates that agree with the bootstrap procedure (min estimate = 40%, max estimate = 44%).

In the case of KS2 and RT1, the estimates from both the bootstrap and smoother procedure were in qualitative agreement with the p-value distributions (*Figure 3—figure supplement 1*). Specifically, a higher proportion of genes downstream of the shared differentially accessible promoter peaks were estimated to be truly differential, and their p-value distributions showed greater concentration close to 0.

Finally, for *Figure 4G*, we employed the analogous procedure to *Figure 3A* in order to estimate the proportion of differentially accessible peaks in promoters of differentially expressed genes for different thresholds.

## Reactome pathway analysis

We used the goseq R package (*Young et al., 2010*) to perform pathway analyses for the shared disrupted genes, based on Reactome pathways (*Jassal et al., 2020*). As our assayed gene set, we used the set of all genes included in all three differential expression analyses, or the set of all genes

that had at least one promoter peak included in all three differential accessibility analysis. As our differential gene set, we used the set of genes differentially expressed in all of the three MDEMs, or the set of genes with at least one differentially accessible promoter peak in all of the three MDEMs. The top 20 enriched pathways are provided in Figure 5 — source data 1 and 2 respectively.

## Transcription factor motifs

We obtained a bed file (mm10.archetype_motifs.v1.0.bed) containing the genomic positions of 233 non-redundant transcription factor motifs, from https://www.vierstra.org/resources/motif_clustering (*Vierstra et al., 2020*). We then restricted to motifs that had at least one base overlapping our set of unionized B-cell peaks (see sections ATAC-seq mapping and peak calling and ATAC-seq differential analysis). Subsequently, we tested each motif for enrichment using the fisher.test() function in R. The differentially accessible peaks at promoters of differentially expressed genes were identified as described in the Identification of differentially expressed genes with differentially accessible promoter peaks section.

## Gene catalogs

### Transcription factors

We obtained a list of 1254 genes encoding for human transcription factors from *Barrera et al., 2016*. We then used the biomaRt R package to obtain the mouse orthologs of these transcription factor genes, with the ENSEMBL IDs as our filter. We only retained high-confidence orthologs ('mmusculus_homolog_orthology_confidence' equal to 1). Finally, we restricted to transcription factors included in all three differential analyses (KS1 vs. WT, KS2 vs. WT, and RT1 vs. WT).

### IgA deficiency genes

We used the Mammalian Phenotype Browser on the Mouse Genome Informatics database (*Leek and Storey, 2007*) to obtain a catalog of genes known to lead to IgA deficiency when individually knocked out. Specifically, we used 'decreased IgA level' as the phenotype term and then obtained all the resulting genes, regardless of the genetic background. In cases of double knockouts, we included both genes.

## ELISA for serum IgA levels

We performed ELISAs on serum IgA from peripheral blood samples as previously described (*Pilarowski et al., 2020*).

## Code availability

All code for the analyses in this manuscript is available at https://github.com/hansenlab/mdem_overlap (copy archived at swh:1:rev:eec9ad39114cf70fd1d313bd99588520a11e7b04, *Leandros, 2021*).

## Acknowledgements

HTB and TRL are supported by a grant from the Louma G Foundation. HTB is also supported by grants from the Icelandic Research Fund (#195835–051, #206806–051) and the Icelandic Technology Development Fund (#2010588–0611). KDH, HTB, and LB were partly supported by the 2016 Discovery Award from Johns Hopkins University. LB was partly supported by the Maryland Genetics, Epidemiology and Medicine (MD-GEM) training program, funded by the Burroughs-Wellcome Fund. Research reported in this publication was supported by the National Institute of General Medical Sciences of the National Institutes of Health under award number R01GM121459. *Figure 1d* was in part created using Biorender; this software has a restrictive license which includes a requirement to be mentioned in the acknowledgements.

## Additional information

### Funding

| Funder | Grant reference number | Author |
|---|---|---|
| National Institutes of Health | R01GM121459 | Kasper Daniel Hansen |
| Icelandic Centre for Research | 195835-051 | Hans T Bjornsson |
| Icelandic Centre for Research | 206806-051 | Hans T Bjornsson |
| Icelandic Centre for Research | 2010588-0611 | Hans T Bjornsson |
| Louma G Private Foundation | KS grant | Teresa R Luperchio Hans T Bjornsson |
| Johns Hopkins University | Discovery grant | Leandros Boukas Kasper Daniel Hansen Hans T Bjornsson |
| Burroughs Wellcome Fund | MD-GEM grant | Leandros Boukas |

The funders had no role in study design, data collection and interpretation, or the decision to submit the work for publication.

### Author contributions

Teresa Romeo Luperchio, Conceptualization, Data curation, Investigation, Methodology, Visualization, Writing – original draft, Writing – review and editing; Leandros Boukas, Conceptualization, Data curation, Formal analysis, Investigation, Methodology, Software, Writing – original draft, Writing – review and editing, Visualization; Li Zhang, Investigation, Methodology; Genay Pilarowski, Jenny Jiang, Allison Kalinousky, Investigation; Kasper D Hansen, Conceptualization, Data curation, Formal analysis, Funding acquisition, Supervision, Visualization, Writing – original draft, Writing – review and editing; Hans T Bjornsson, Conceptualization, Funding acquisition, Investigation, Project administration, Resources, Supervision, Writing – original draft, Writing – review and editing, Visualization

### Author ORCIDs

Teresa Romeo Luperchio http://orcid.org/0000-0001-8586-3101
Leandros Boukas http://orcid.org/0000-0002-6464-867X
Jenny Jiang http://orcid.org/0000-0002-8310-298X
Allison Kalinousky http://orcid.org/0000-0001-7292-8500
Hans T Bjornsson http://orcid.org/0000-0001-6635-6753

### Ethics

We performed all mouse experiments in accordance with the National Institutes of Health Guide for the Care and Use of Laboratory Animals and all were approved by the Animal Care and Use Committee of the Johns Hopkins University. (protocol number: MO18M112).

### Decision letter and Author response

Decision letter https://doi.org/10.7554/eLife.65884.sa1
Author response https://doi.org/10.7554/eLife.65884.sa2

## Additional files

### Supplementary files

• Transparent reporting form

### Data availability

Sequencing data have been deposited in GEO under accession code GSE162181.

The following dataset was generated:

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
