## [Decision Letter]

**Acceptance summary:**

This manuscript finds common molecular features in the blood of three mouse models for neurodevelopmental disorders, Kabuki syndromes 1 and 2 and Rubinstein-Taybi syndrome type 1. These shared features (chromatin accessibility and gene expression) may underlie some of the clinical similarities of these disorders. This work will be of interest to researchers, and to some clinicians studying neurodevelopmental disorders.

**Decision letter after peer review:**

Thank you for submitting your article "Leveraging the Mendelian Disorders of the Epigenetic Machinery to Systematically Map Functional Epigenetic Variation" for consideration by *eLife*. Your article has been reviewed by 2 peer reviewers, and the evaluation has been overseen by a Reviewing Editor and Huda Zoghbi as the Senior Editor. The reviewers have opted to remain anonymous.

The reviewers have discussed their reviews with one another, and the Reviewing Editor has drafted this to help you prepare a revised submission. The reviewers and the Reviewing Editor find that an extensive revision is required for the work to be suitable for publication in *eLife*.

Essential revisions:

1) The introduction is rather short, although the journal does not limit the length of this section. For instance, there is no introduction to the disease phenotypes, what genes are implicated in these diseases, and what is known about their clinical and molecular overlap. As some of this information is in the first sections of the results, the authors should consider moving these paragraphs to the introduction.

2) The authors' rationale for focusing their studies on these conditions is the overlapping immune system dysfunction in these disorders. This overlap should be better described. As immune dysfunction may be biased with respect to sex this issue needs to be addressed as well.

3) The use of ATAC- and RNA-seq data from the same mice as well as the exploration of the GO function of these genes is an important contribution to the literature. The hypothesis of the study (as stated and shown in figure 1) is that shared molecular (ATAC-seq peaks) features across the three conditions cause the common phenotypic features of these conditions. While stating the hypothesis in this way clearly frames the motivation of the study, it is not strictly tested by the experiments performed. While this study moves our understanding of these disorders forward, it does not directly address causation. Further, the GO functions and TFs identified are not clearly linked to disorder-relevant pathways. The authors should consider reframing the hypothesis to better reflect the advances presented in the paper.

4) The ATAC-seq data are generally clearly presented, but there are a few areas where this could be improved:

– Line 139: it would be beneficial to the reader to give a quick summary of the new method to detect overlapping peaks here, so they are not required to flip to the methods section to understand the main idea.

– The authors did not indicate the rationale behind their choice of performing the experiments in 2.5-to 3.5 old mice.

– In the fast ATAC-seq protocol, the authors need to clarify the number of mice used for each comparison -maybe best to add them to Figure 1 and also indicate that all are female which is a limitation of the study.

– What is the justification for using a 10% FDR over a more standard 5%?

– 10% may be justified given that the analysis involves overlapping multiple datasets, and a more permissive q-value cut-off was used to capture more potential findings. If this is the justification, it should be noted.

– In figure 2, why is (f) not shown on the same plot as e)? It appears that the shared peaks differentiate KS1 and KS2, (and perhaps K2S and KS1 from RTS if plotted together). This would seem to be a very notable finding, that is the "shared" peaks differ in the magnitude of change for each condition. Can the authors clarify these data especially from a quantitative perspective, and elaborate on their interpretation?

In the methods, the authors stated that they counted the number of reads that map to each of the 78,193 features (overlapping DHS) from each sample and they only included features with median (across samples) counts greater than 10. How many of these 78,193 features are left for differential analysis? In addition, the authors state that they used surrogate variable analysis to estimate unobserved confounding variables which they adjusted for. Can the authors clearly describe the confounding variables that were identified?

– The PCA on the 313 overlapping promoter peaks (of the initial 824 identified in KS1) within the 3 disorders separate KS1 from KS2 (Figure 2e) , The authors state that this is surprising given the clinical overlap but the authors have only used female KS2 animals. As this is an X-linked gene, these mice may not necessarily exhibit the full KS2 phenotypic spectrum. The rationale for choosing female mice exclusively needs to be provided and the interpretation of the data should be contextualized for only female data.

5) The RNA-seq analysis needs to be improved:

– The authors state that a subset of the mice from the ATAC-seq experiment were used for RNA-seq, can they clarify the exact nature of the overlap? Were all RNA-seq mice part of the ATAC-seq experiment?

– Again, Figure 4 (e) and (f) are not plotted together, making interpreting the nature of the relationships across disorders (a primary outcome of this study) more difficult. And here again, it appears that though many gene expression changes are shared, the magnitude of the change differentiates the conditions from each other. If this is the case can the authors elaborate-are the changes/differences statistically significant?

– Given the low adjusted p-value threshold, have the authors attempted to validate a subset of their findings using another method such as qPCR?

6) In the discussion, (line 330) I agree that the functional relevance of DNA methylation signatures is unclear at this time, due to blood being peripheral to brain. However, several studies have found indications that they may be indeed be relevant. For DNA methylation data, cell type-heterogeneity is addressed better in blood than many other tissues, and is seldom ignored in studies published in the last 2-3 years. When drawing parallels to DNA methylation studies, I would suggest the authors highlight the much stronger relationship between ATAC-seq and gene expression, than that for DNA methylation and gene expression (as demonstrated in this study).

7. The authors need to examine further their methodology to see if they can come to a more definitive understanding of what is driving the predominant skew to open chromatin and the lack of enriched targets for these specific chromatin modifier mutations. Instead of their grouped alignment for their ATAC-seq, they should align the samples individually and examine the previously identified loci for any indication of differences between the samples that may indicate technical confounding, or else. As mentioned, rare genetic variation can still occur in substrains (Watkins-Chow and Pavan, PMID: 18032724), so this should be excluded via SNPs, CNV, Indels within C57BL/6J from Mortazavi et al., (BioRxiv 10.1101/2020.03.16.993683), other datasets, or explored more directly in their data. Although clearly more predominate in human populations, common or rare allelic genetic variation via SNP/ CNV/Indel/Polymorphic Repeats within promoters in loci can drive epigenetic/open chromatin variation (Leinert et al., PMID: 21964573; Do et al., PMID27153397; Martin-Trujillo et al., PMID: 33216750). Although the authors found their loci were predominately within DHSs, this does not preclude that some of these DHS may be polymorphic. One possibility is that the grouping may be accentuating open regions in the potentially more homogeneous mutant strains in comparison to the wild-type mice. If potential strain variation background is adding noise to real results, these may then become more apparent if these are removed, enabling more explainable pathway effects that can be easier interpreted and bring the insights aimed for.

8. The authors state that these Mendelian trait monogenic diseases should be regarded as complex traits "arising from widely distributed epigenetic perturbations across the genome". This may, however, lead to some confusion regarding genetic terminology as they are Mendelian disorders. Instead, the authors could put this in the context of the expanding understanding of a complex modifier landscape of underlying genetic background effects as well as environmental and stochastic factors on penetrance and expressivity of monogenic traits (Oetjens et al., PMID: 31653860; Wright et al., PMID: 30665703; Czyz et al. PMID: 22898292).

9. The authors highlight the overlapping phenotypic features of intellectual disability, growth defects, and immune dysfunction as the justification for looking at these d,ifferent primary genetic defects for common downstream epigenomic alterations. However, they should also acknowledge that specific analysis of each MDEM may bring a more precise evaluation of the disease-driving signals, as multiple distinct pathways can lead to these very broad phenotypes.

10. In regard to the examples in the Introduction of robust relationships between epigenetic alterations and specific phenotypes – the authors should also include the oncogenic role of the promoter methylation of mismatch repair gene MLH1 (Gazzoli et al., PMID:12124320). Additionally, there is no current strong evidence of a role of 'metastable epialleles' in human phenotypes.

11. Authors should also acknowledge that some aspects of the human immunological features of these MDEMs will not be directly comparable in mice (Seok et al., PMID: 23401516, etc).

12. The comment in the Discussion, re DNA methylation 'epi-signatures' should also take into account that these are being used as 'biomarkers' to help improve diagnosis of variants of indeterminate significance in monogenic traits. Therefore, they themselves do not all have to be proved functional to have clinical utility. Furthermore, that this also takes advantage of the fact that the DNA methylome and chromatin modifications are highly interconnected (Thomson et al., PMID: 20393567; Baubec et al., PMID: 25607372, etc).

---

## [Author Response]

Essential revisions:1) The introduction is rather short, although the journal does not limit the length of this section. For instance, there is no introduction to the disease phenotypes, what genes are implicated in these diseases, and what is known about their clinical and molecular overlap. As some of this information is in the first sections of the results, the authors should consider moving these paragraphs to the introduction.

We have now restructured the introduction/first Results section to address these points. Specifically, we moved three out of the four paragraphs that were previously in the first Results section to the introduction. In addition, we have now expanded our description of the disease phenotypes, with a focus on the immune phenotypes and their overlap (see also our response to comment 2). The first Results section now only contains a single paragraph, where we give an overview of the new statistical approach, we developed for the overlap analysis. We have added subheadings to the introduction, as we believe it helps to separate the description of the general problem we set out to address (mapping functional disease-associated epigenetic variation), from the description of the three specific disorders that we used as proof-of-principle, and their immune phenotypes.

2) The authors' rationale for focusing their studies on these conditions is the overlapping immune system dysfunction in these disorders. This overlap should be better described.

We have now included an expanded description of the immune phenotypes in each disorder and their overlap in our introduction (see also response to previous comment).

We now have text that stands as follows:

“The shared phenotypes of these three syndromes include intellectual disability, growth retardation, and immune dysfunction; the latter is our focus here. In KS1, the immune dysfunction includes hypogammaglobulinemia, with low IgA as a consistent feature, as well as abnormal cell maturation which has mostly been characterized in B cells^12,20,21^. RT1 can also manifest with hypogammaglobulinemia and reduction of mature B cells^22^. These defects in KS1 and RT1 are thought to (at least partly) explain the increased susceptibility to infections. In KS2, the immune phenotype has been less extensively studied, in part due to the rarity of the disorder, but there is some evidence of increased infection susceptibility, and hypogammaglobulinemia^20,23^. In mice, the immune phenotypes have been studied in depth only in KS1, and the IgA deficiency phenotype closely resembles what is seen in patients.”

As immune dysfunction may be biased with respect to sex this issue needs to be addressed as well.

We have now acknowledged this in our discussion. It is also worth mentioning that, in Kabuki syndrome type 1, previous results suggest that the immune defects are similar in both male and female patients, and male and female mice (current citation 17). This suggests the possibility that our results here are relevant to males as well. Of course, further work is needed to test this in KS2 and RT1. In the second paragraph of the discussion, we now write:

“One limitation in our study is the use of only female mice. While it is known that sex differences can influence immune responses and deficiencies^29^, IgA deficiency has been observed in both male and female KS1 patients, and characterized in both male and female KS1 mice^17^. This supports the notion that our results are relevant to both sexes, although future work is needed to test this for KS2 and RT1.”

In addition, in the introduction and in the methods, we now explain that Kdm6a haploinsufficiency is lethal in the KS2 model, necessitating a female only comparison. In the last paragraph of the introduction, we now write:

“In order to facilitate a direct comparison of the three MDEMs, we only used female mice, as Kdm6a is on the X chromosome (KS2 mouse model) and its complete loss (full knockout) is lethal in male mice.”

And in the “Sex disaggregation” section of the methods, we write:

“We performed all experiments in female mice to enable comparison between all three disease models, as Kdm6a (KS2 model) is present on the X chromosome, and its full knockout is in our experience uniformly lethal in male mice. Therefore, we are unable to present sex-disaggregated data.”

3) The use of ATAC- and RNA-seq data from the same mice as well as the exploration of the GO function of these genes is an important contribution to the literature. The hypothesis of the study (as stated and shown in figure 1) is that shared molecular (ATAC-seq peaks) features across the three conditions cause the common phenotypic features of these conditions. While stating the hypothesis in this way clearly frames the motivation of the study, it is not strictly tested by the experiments performed. While this study moves our understanding of these disorders forward, it does not directly address causation. Further, the GO functions and TFs identified are not clearly linked to disorder-relevant pathways. The authors should consider reframing the hypothesis to better reflect the advances presented in the paper.

Our hypothesis has two components: (1) that there exist shared abnormalities between MDEMs at the molecular (epigenomic/transcriptomic) level, and (2) that these shared molecular abnormalities play a causal role in the pathogenesis of the shared disease manifestations. With the experiments performed here, we took a first step towards validating our hypothesis by testing the first component, and identifying loci with disrupted accessibility, and genes with disrupted expression, in all three MDEMs studied. We fully recognize that future work is required to definitively prove that these

shared loci/genes are causally involved in disease pathogenesis (component 2). We have now rewritten the first paragraph of the discussion to explicitly state this. We write:

“Our study is motivated by the hypothesis that the shared phenotypic manifestations seen in MDEMs are attributable to shared underlying epigenomic and transcriptomic abnormalities. We have taken a first step towards validating this hypothesis, by showing that three MDEMs caused by loss-of-function variants in three distinct epigenetic regulators have shared alterations at the chromatin and gene expression level in B cells. However, we have not established that these shared abnormalities are causal for pathogenesis; this will require targeted manipulation of chromatin state and gene expression at the appropriate cell types and developmental stages.”

Regarding the Reactome pathway analysis, we believe the fact that we do not observe “smoking guns”, but rather several pathways appear dysregulated, does not in any way run contrary to our hypothesis. It merely shows that the shared abnormalities are not confined to a small discrete set of genes/pathways, but rather reflect a more widespread perturbation of multiple cellular processes. Of course, whether this interpretation is correct is something that will be definitively proven in future work. We also want to point out that our results on the collective dysregulation of IgA-relevant genes and TF genes, while consistent with an underlying systems-level dysregulation, do provide links to the specific phenotypic manifestations of IgA deficiency and abnormal B cell maturation, respectively. We have now modified the relevant parts of the discussion to incorporate these points. In the final sentence of the first discussion paragraph, we write:

“Nonetheless, our results provide some evidence of functionality, as illustrated by the fact that: (a) many chromatin changes at promoters are linked to downstream gene expression changes, and (b) systematic expression changes affect genes known to contribute to specific, well-characterized phenotypic features (IgA deficiency, abnormal B cell maturation) of these MDEMs.”

And in the first half of the third discussion paragraph, we write:

“In terms of understanding the pathogenesis of MDEMs, our results clearly point towards a generalized, systems-level dysregulation, with a multitude of cellular processes/pathways affected. This is supported both by the extensive sharing of chromatin and expression alterations between the three disorders, as well as by the several Reactome pathways that appear affected. From our present study it is unclear how exactly these combine to ultimately give rise to the phenotypic manifestations; elucidating this will be an important challenge going forward.”

4) The ATAC-seq data are generally clearly presented, but there are a few areas where this could be improved:– Line 139: it would be beneficial to the reader to give a quick summary of the new method to detect overlapping peaks here, so they are not required to flip to the methods section to understand the main idea.

We now write:

“Briefly, we first obtained the 1,062 differential promoter peaks from the KS1 vs WT analysis. Then, we used the distribution of the p-values for these peaks from the KS2 vs WT analysis to estimate the percentage of shared differential peaks, and individually label each peak as shared or not at the 10% FDR threshold (see Methods for details).”

– The authors did not indicate the rationale behind their choice of performing the experiments in 2.5-to 3.5 old mice.

Our rationale was that this is the age range when we understand most about the KS1 mice, and also is when the IgA deficiency appears. We now state this in the results, right before going into the data. We write:

“We chose to limit the age range to 2.5-3.5 months as this is the age range we know most about this KS1 mode, and this is when the IgA deficiency first manifests in KS1 mice^17^.”

– In the fast ATAC-seq protocol, the authors need to clarify the number of mice used for each comparison -maybe best to add them to Figure 1 and also indicate that all are female which is a limitation of the study.

We now state the number of mice used when describing each comparison in the results section, and we have also added a table with the sample size of our experiments as a new panel in Figure 1 (current panel E). In addition, we now discuss the use of female mice at several places in our manuscript (see comments above and the comment on the PCA plot below).

– What is the justification for using a 10% FDR over a more standard 5%?– 10% may be justified given that the analysis involves overlapping multiple datasets, and a more permissive q-value cut-off was used to capture more potential findings. If this is the justification, it should be noted.

Indeed, since the primary goal of our study is to identify differentially accessible loci/differentially expressed genes shared between the disorders, we decided that a less stringent significance threshold is preferable, as each differential analysis can be viewed as a kind of “replication”. It is also worth noting here that the FDR threshold adopted only affects the identification of specific loci/genes that belong to the overlap between the disorders; it does not affect our estimates of the percentage of shared loci/genes, since these are derived from the distribution of all relevant p-values. In addition, while a 5% cutoff is standard when testing a single null hypothesis, we respectfully disagree that 5% is standard when assessing multiple null hypotheses by controlling the false discovery rate (FDR). For example, in the RNA-seq analysis in our case, using an FDR of 10% amounts to expecting that, on average, 10% of the genes we label as shared differential are going to be false positives. If we assume that the realized FDR in our particular experiment is close to the expectation of 10%, this means that approximately 26 out of the 264 genes we discovered as differentially expressed in all three disorders will be false positives. We consider this to be an acceptable rate of false positives, and it is our impression that this is shared by a sizeable number of researchers in genomics. That said, we fully recognize that the false positive rate that an investigator feels comfortable with is, at the end of the day, subjective. For that reason, in the tables that we provide with the differentially accessible peaks/differentially expressed genes, we have now also added an extra column specifying whether each peak/gene passes the 0.05 significance threshold, so someone who uses our data can adopt this more stringent threshold if they prefer so.

– In figure 2, why is (f) not shown on the same plot as e)? It appears that the shared peaks differentiate KS1 and KS2, (and perhaps K2S and KS1 from RTS if plotted together). This would seem to be a very notable finding, that is the "shared" peaks differ in the magnitude of change for each condition. Can the authors clarify these data especially from a quantitative perspective, and elaborate on their interpretation?

As suggested by the reviewer, we have now replaced main figures 2e and 2f with a

single joint PCA plot (same for 4e and 4f for the RNA-seq PCA plots; see comment below). Our rationale for initially choosing to separately perform the principal component analysis for KS1 and KS2 vs wild-type mice, and RT1 vs wild-type, was that these were two separate experiments in terms of cell isolation, library preparation, and genetics (the wild-type mice are sibs/cousins of the mutants). We thus expected that the batch effect would be the dominant source of variation if the PCA was performed jointly for all mice. However, we have now investigated this, and we observe that, using the shared differentially accessible promoter peaks (as identified with our method), the largest source of variation in the joint PCA plot is mutant/wild-type status. We have kept the original separate PCA plots as supplemental figures. It is also worth noting that this result, namely that the variation from the batch effect is less than the variation from the mutant/wild-type status, even when combining two separate experiments, provides additional support to the robustness of our shared differential analysis results.

Now, in the joint PCA plot using the shared promoter ATAC peaks, the KS2 mice cluster in-between the KS1 and the wild-type mice (as was the case originally in the separate plot), while the RT1 mice cluster closer to the KS1. This suggests that, as the reviewer states, the accessibility alterations have a smaller effect size in KS2, something that is mirrored in the RNA-seq PCA plot as well. We agree that this is potentially very interesting, and we now comment on it in the results. We also raise the possibility that this is because we are using female mice, which are expected to be mosaic with respect to the Kdm6a knockout since Kdm6a is on the X chromosome.

We write:

“A principal component analysis shows that the accessibility signal of these shared disrupted promoter peaks separates each of the three mutant genotypes from their wildtype littermates (Figure 2e; Supplemental Figure 2). The KS1 mice cluster close to RT1, while KS1 and KS2 cluster separately from each other, with KS2 being closer than KS1 to wild-type, indicating smaller effect sizes of the accessibility alterations. This KS1/2 separation is surprising, since patients have such strong phenotypic overlap that the two syndromes were not considered distinct prior to discovery of the causative genes. However, it should be noted here that, since Kdm6a is on the X chromosome, the KS2 female mice are expected to be mosaic with respect to Kdm6a knockout, and this may explain the smaller magnitude of their accessibility defects.”

In the methods, the authors stated that they counted the number of reads that map to each of the 78,193 features (overlapping DHS) from each sample and they only included features with median (across samples) counts greater than 10. How many of these 78,193 features are left for differential analysis?

We now provide this information in the methods section, both for the ATAC-seq and RNA-seq analyses. For ATAC-seq, we write:

“We only retained features with a median (across samples) count greater than 10; this filter resulted in 71,651 features for the KS1 vs WT analysis, 73,189 features for the KS2 vs WT analysis, and 62,386 features for the RT1 vs WT analysis.”

And for RNA-seq:

“The exclusion of genes with median count across samples less than or equal to 10 resulted in 12,566 genes tested in the KS1 vs WT analysis, 12,529 genes tested in the KS2 vs WT analysis, and 12,537 genes tested in the RT1 vs WT analysis.”

In addition, the authors state that they used surrogate variable analysis to estimate unobserved confounding variables which they adjusted for. Can the authors clearly describe the confounding variables that were identified?

We investigated whether the surrogate variables identified using SVA in our differential analyses have a clear association with variables we would expect to act as confounders in the differential analysis. As possible confounders we investigated cohort and day of bleeding/cell isolation, and only found a very mild correlation with the SVs in the ATACseq analysis. It is worth noting that this is not at all uncommon. While in some cases the SVs correlate highly with known confounding variables, that is certainly not always the case, and this is an advantage of SVA. Loosely speaking, this is explained by the fact that SVA does not attempt to directly estimate the unobserved confounding variables; what it estimates is a set of orthogonal vectors which span the same vector space as the unobserved confounders.

For completeness, we now provide the surrogate variables identified in all our analysis (ATAC- and RNA-seq) as supplemental tables.

– The PCA on the 313 overlapping promoter peaks (of the initial 824 identified in KS1) within the 3 disorders separate KS1 from KS2 (Figure 2e) , The authors state that this is surprising given the clinical overlap but the authors have only used female KS2 animals. As this is an X-linked gene, these mice may not necessarily exhibit the full KS2 phenotypic spectrum. The rationale for choosing female mice exclusively needs to be provided and the interpretation of the data should be contextualized for only female data.

See also our response to point number 3 above. The reason for using female mice only is that Kdm6a haploinsufficiency is male lethal in our mouse model. We now explicitly state this in the last paragraph of the introduction, where we write:

“In order to facilitate a direct comparison of the three MDEMs, we only used female mice, as Kdm6a is on the X chromosome (KS2 mouse model) and its complete loss (full knockout) is lethal in male mice.”

And in the “Sex disaggregation” section of the methods, where we write:

“We performed all experiments in female mice to enable comparison between all three disease models, as Kdm6a (KS2 model) is present on the X chromosome, and its full knockout is in our experience uniformly lethal in male mice. Therefore, we are unable to present sex-disaggregated data.”

With regards to the PCA plot, we agree that one potential reason for the separation between KS1 and KS2 is the fact that we are using female mice. Therefore, due to the fact that Kdm6a is on the X chromosome, the KS2 mice are expected to be mosaic with respect to the Kdm6a knockout, which may lead to a milder phenotype. We have now included a comment raising this possibility in the results, where we write:

“A principal component analysis shows that the accessibility signal of these shared disrupted promoter peaks separates each of the three mutant genotypes from their wildtype littermates (Figure 2e; Supplemental Figure 2). In addition, KS1 and KS2 cluster separately from each other, with KS2 being closer to wild-type, indicating smaller effect sizes of the accessibility alterations. This KS1/2 separation is surprising, since patients have such strong phenotypic overlap that the two syndromes were not considered distinct prior to discovery of the causative genes. However, it should be noted here that, since Kdm6a is on the X chromosome, the KS2 female mice are expected to be mosaic with respect to Kdm6a knockout, and this may explain the smaller magnitude of their accessibility defects.”

5) The RNA-seq analysis needs to be improved:– The authors state that a subset of the mice from the ATAC-seq experiment were used for RNA-seq, can they clarify the exact nature of the overlap? Were all RNA-seq mice part of the ATAC-seq experiment?

Indeed, all mice from the RNA-seq data were part of the ATAC-seq experiment also. Mice were bled, cells were isolated and then aliquoted for ATAC or RNA processing, with the RNA-seq samples being an aliquot of the same cell harvest as the ATAC-seq samples. We have edited the text to clarify this, both in the Methods and in the Results.

In the Methods, we now write:

“We isolated CD19+ B cells by positive selection using CD19+ microbeads for mouse (Miltenyi 130-052-201) following manufacturer protocols, then counted and aliquoted samples on ice to further process for ATAC-seq and RNA-seq. We note that each RNAseq sample was an aliquot of the same cell harvest as the ATAC-seq sample from the same mouse (though for some mice we only performed ATAC-seq and not RNA-seq).”

And in the Results, we write:

“To capture both chromatin and transcriptional status at a single time point, we generated the RNA-seq samples in parallel with the samples used for ATAC-seq, from a subset of the same individual mice (Methods). Specifically, we performed RNA-seq on 5 KS1 mice, 5 KS2 mice, 5 RT1 mice, and 5 and 7 wild-type mice from the Kabuki and Rubinstein-Taybi cohorts, respectively.”

– Again, Figure 4 (e) and (f) are not plotted together, making interpreting the nature of the relationships across disorders (a primary outcome of this study) more difficult. And here again, it appears that though many gene expression changes are shared, the magnitude of the change differentiates the conditions from each other. If this is the case can the authors elaborate-are the changes/differences statistically significant?

Following the reviewer’s suggestion, we have now replaced figures 4e and f with a single joint PCA plot for all mice. As also explained in our response to the comment about the ATAC-seq PCA plot, our rationale for originally presenting these two PCA plots separately was that they represent two separate experiments with regards to cell isolation, library preparation and mouse genetics. This led us to assume that the dominant source of variation, if the PCA was performed jointly on all mice, would be the batch effect. However, after investigating this we found that, as with the ATAC-seq PCA plot, the major source of variation in the data when using the shared differentially expressed genes is mutant/wild-type status. As with ATAC-seq, we have kept the original separate PCA plots as supplemental figures. Regarding the effect size, it indeed appears that the KS2 mice have changes of smaller magnitude at the shared differentially expressed genes. This mirrors what is seen at the shared differentially accessible ATAC-seq promoter peaks. One potential explanation for this, which we have now included in our manuscript based on the reviewer’s previous suggestion, is mosaicism, because Kdm6a is on the X chromosome. Finally, all changes are statistically significant, but note that since our statistical approach is designed to increase power for the overlap analysis, some of the statistically significant shared changes do not come up as statistically significant individually.

– Given the low adjusted p-value threshold, have the authors attempted to validate a subset of their findings using another method such as qPCR?

We chose not to validate individual expression changes by qPCR since we felt that this would not add much value to current data, as it would not be entirely complementary/independent to what is learned by RNA-seq. However, it is worth mentioning that (1) our study is better powered than many other mouse studiesperforming ATAC-seq and RNA-seq, (2) one could also think of individual disorders as a type of replication and (3) ATAC-Seq and RNA-Seq data complement each other (and support our findings) yet are based on very different techniques with different biases.

6) In the discussion, (line 330) I agree that the functional relevance of DNA methylation signatures is unclear at this time, due to blood being peripheral to brain. However, several studies have found indications that they may be indeed be relevant. For DNA methylation data, cell type-heterogeneity is addressed better in blood than many other tissues, and is seldom ignored in studies published in the last 2-3 years. When drawing parallels to DNA methylation studies, I would suggest the authors highlight the much stronger relationship between ATAC-seq and gene expression, than that for DNA methylation and gene expression (as demonstrated in this study).

We have now rewritten that part of the discussion in order to more accurately highlight the differences between our approach and that of studies deriving DNA methylation signatures. Upon reconsideration, (1) we no longer mention the issue of cell-type heterogeneity, and (2) we directly acknowledge that methylation signatures may be functionally relevant. We have decided to not contrast ATAC-seq and DNA methylation data with regards to their relationship with gene expression, as we do not have DNAm measurements (a direct comparison of ATAC vs DNAm can be found in Rizzardi et al., (2019) Nature Neuroscience).

We now write:

“We note that our study differs from recent studies of DNA methylation in the peripheral blood of MDEM patients^35-37^. In these studies, the goal is to derive “episignatures” with the capacity for robust phenotypic prediction. As a result, these episignatures include a set of CpGs that jointly maximize the ability to separately classify individuals with a given MDEM from controls, without regard to the causal role (if any) of these CpGs in disease pathogenesis. While this does not limit their potential usefulness, and it does not exclude the possibility that changes in the methylation state of some of these CpGs may be functionally related to disease pathogenesis, our strategy is specifically designed to yield a catalog of abnormalities with primary functional role in shared MDEM pathogenesis.”

7. The authors need to examine further their methodology to see if they can come to a more definitive understanding of what is driving the predominant skew to open chromatin and the lack of enriched targets for these specific chromatin modifier mutations. Instead of their grouped alignment for their ATAC-seq, they should align the samples individually and examine the previously identified loci for any indication of differences between the samples that may indicate technical confounding, or else. As mentioned, rare genetic variation can still occur in substrains (Watkins-Chow and Pavan, PMID: 18032724), so this should be excluded via SNPs, CNV, Indels within C57BL/6J from Mortazavi et al., (BioRxiv 10.1101/2020.03.16.993683), other datasets, or explored more directly in their data. Although clearly more predominate in human populations, common or rare allelic genetic variation via SNP/ CNV/Indel/Polymorphic Repeats within promoters in loci can drive epigenetic/open chromatin variation (Leinert et al., PMID: 21964573; Do et al., PMID27153397; Martin-Trujillo et al., PMID: 33216750). Although the authors found their loci were predominately within DHSs, this does not preclude that some of these DHS may be polymorphic. One possibility is that the grouping may be accentuating open regions in the potentially more homogeneous mutant strains in comparison to the wild-type mice. If potential strain variation background is adding noise to real results, these may then become more apparent if these are removed, enabling more explainable pathway effects that can be easier interpreted and bring the insights aimed for.

This question is fully addressed by our experimental design, which we believe is the most appropriate experimental design for our study. Since it appears that we have not communicated this effectively in our manuscript, we have taken steps to improve our exposition (see below).

In detail: the phenotypes we are studying are caused by a single loss-of-function allele, ie. the mutant mice carry a heterozygous loss-of-function variant. All three mouse models have been backcrossed onto C57BL/6J in our laboratory. For this study, multiple mating pairs of wild-type crossed with heterozygous mutants were used. The resulting litters from these mate pairs were a mix of mutant and wild-type offspring, all sharing the same genetic background. The wild-type controls that we use in our differential analyses are therefore either siblings to the mutants, or cousins (because we have many litters of mice) within each disease model cohort. This design ensures that the confounding by genetic variation due to background differences which the reviewers describe does not affect our study.

To better communicate these points in our manuscript, we write in our methods section

“Mice””: “KS1. Kmt2d^+/βGeo^ mice are fully backcrossed onto C57BL/6J and this backcrossing is verified by SNP genotyping^38^. These mice are also known as Mll2^Gt(RRt024)Byg^. They were originally obtained from BayGenomics and were fully backcrossed in the Bjornsson laboratory.”

And

“KS2. Kdm6a+/- mice were acquired from European Mouse Mutant Archive (EMMA) but this model has also been called: Kdm6a^tm1a(EUCOMM)Wtsi^. Mice were crossed with flippase expressing mice (B6.Cg-Tg(ACTFLPe)9205Dym/J) (Jackson Laboratories) to remove the third exon of Kdm6a, and then progeny were crossed with Cre expressing mice driven by CMV (B6.C-Tg(CMV-cre)1Cgn/J) (Jackson Laboratories), to generate the Kdm6a^tm1d(EUCOMM)Wtsi^ allele. Mice were backcrossed onto C57BL/6J to maintain the Kdm6^atm1d(EUCOMM)Wtsi^ allele.”

And

“RT1. Crebbp+/- mice also known as Crebbp^tm1Dli^, were acquired from Jackson laboratories but established by Kung et al^39^. These mice were maintained on a C57BL/6J background in the Bjornsson laboratory.”

And

“For this study, multiple mating pairs of wild-type mice crossed with heterozygous mutants were used. The resulting litters from these mate pairs were a mix of mutant and wild-type offspring, all sharing the same genetic background. The wild-type controls that we use in our differential analyses are therefore either siblings to the mutants, or cousins (because we have many litters of mice) within each disease model cohort. This ensures our study is not confounded by systematic differences in genetic background between wild-type and mutants.”

Finally, the last sentence of the comment touches upon the desire to have clear, easily explainable pathways. It is true that this is not what we have found. However, as also explained in our response to a previous comment, and in the response to the comment that follows, we do not consider this a limitation. Instead, we view it as one of the contributions of our study that we have performed a careful assessment, which indicates that MDEMs (at least the three MDEMs we have studied here) seem to have subtle, but widespread, alterations throughout the genome. We realize this is conceptually different from the way we usually think about single-gene disorders (see also comment below), and we have had long internal discussions about this issue, but we believe it is the correct interpretation of the data.

8. The authors state that these Mendelian trait monogenic diseases should be regarded as complex traits "arising from widely distributed epigenetic perturbations across the genome". This may, however, lead to some confusion regarding genetic terminology as they are Mendelian disorders. Instead, the authors could put this in the context of the expanding understanding of a complex modifier landscape of underlying genetic background effects as well as environmental and stochastic factors on penetrance and expressivity of monogenic traits (Oetjens et al., PMID: 31653860; Wright et al., PMID: 30665703; Czyz et al. PMID: 22898292).

We have now modified this sentence in order to better clarify our point and avoid confusion. We believe that the key issue here is the trans-acting nature of the causative epigenetic regulators, which normally bind to many genomic locations. Because of this, haploinsufficiency for these factors has widespread effects across the genome. Our results support the notion that even though many of these widespread effects are individually subtle, collectively they contribute to the phenotypes. Therefore, we do not think that the most relevant explanation involves modifier effects or environmental/stochastic factors, since these are pertinent to issues such as penetrance/expressivity (as the reviewer states). Rather, the distinction we want to make is between the mode of inheritance (Mendelian) and the underlying molecular pathogenesis (resembles what is seen in complex disorders). We believe our modified language now makes this clear. We write:

“It is also worth noting that the emergent picture bears similarities to the molecular basis of complex diseases. This is perhaps not unexpected, given that epigenetic regulators are typically trans-acting proteins that act at many locations. It also suggests that, even though MDEMs are single-gene Mendelian disorders with respect to their inheritance pattern, when it comes to their underlying molecular pathogenesis they might best be conceptualized as effectively complex disorders, with many widely distributed, smalleffect perturbations, ultimately generating the phenotype^31^.”

9. The authors highlight the overlapping phenotypic features of intellectual disability, growth defects, and immune dysfunction as the justification for looking at these different primary genetic defects for common downstream epigenomic alterations. However, they should also acknowledge that specific analysis of each MDEM may bring a more precise evaluation of the disease-driving signals, as multiple distinct pathways can lead to these very broad phenotypes.

We have now added a sentence in the Discussion acknowledging this. We write:

“We also note that, for a complete understanding of each MDEM individually, our cross- MDEM comparison approach should ultimately be complemented by disorder-specific analyses, as some disrupted loci/genes/pathways may show disorder-specific abnormalities.”

10. In regard to the examples in the Introduction of robust relationships between epigenetic alterations and specific phenotypes – the authors should also include the oncogenic role of the promoter methylation of mismatch repair gene MLH1 (Gazzoli et al., PMID:12124320). Additionally, there is no current strong evidence of a role of 'metastable epialleles' in human phenotypes.

We have now added a sentence in the Discussion acknowledging this. We write:

“We also note that, for a complete understanding of each MDEM individually, our cross-MDEM comparison approach should ultimately be complemented by disorder-specific analyses, as some disrupted loci/genes/pathways may show disorder-specific abnormalities.”

11. Authors should also acknowledge that some aspects of the human immunological features of these MDEMs will not be directly comparable in mice (Seok et al., PMID: 23401516, etc).

We have acknowledged this in the second paragraph of our discussion, where we write:

“It should also be mentioned that, given the differences in immune system function between mice and humans^30^, some aspects of our results (e.g. some of the disrupted loci/genes) may differ in patients. However, the immune dysfunction in KS1 mice has previously been shown to mimic many aspects of what is seen in patients^17^, and we therefore anticipate that a substantial proportion of the specific changes will be recapitulated. We also expect that the pattern of extensive sharing of abnormalities between MDEMs will hold true.”

12. The comment in the Discussion, re DNA methylation 'epi-signatures' should also take into account that these are being used as 'biomarkers' to help improve diagnosis of variants of indeterminate significance in monogenic traits. Therefore, they themselves do not all have to be proved functional to have clinical utility. Furthermore, that this also takes advantage of the fact that the DNA methylome and chromatin modifications are highly interconnected (Thomson et al., PMID: 20393567; Baubec et al., PMID: 25607372, etc).

We have rewritten this segment of the discussion (see also our response to comment above). We recognize that DNA methylation signatures serve a diagnostic purpose, and therefore do not have to be functionally relevant to disease pathogenesis in order to be useful in the clinical setting. In addition, we have now acknowledged that some of these CpGs may indeed be functional. We now write:

“We note that our study differs from recent studies of DNA methylation in the peripheral blood of MDEM patients^35-37^. In these studies, the goal is to derive “episignatures” with the capacity for robust phenotypic prediction. As a result, these episignatures include a set of CpGs that jointly maximize the ability to separately classify individuals with a given MDEM from controls, without regard to the causal role (if any) of these CpGs in disease pathogenesis. While this does not limit their potential usefulness, and it does not exclude the possibility that changes in the methylation state of some of these CpGs may be functionally related to disease pathogenesis, our strategy is specifically designed to yield a catalog of abnormalities with primary functional role in shared MDEM pathogenesis.”